# Spatiotemporal dynamics of fast electron heating in solid-density matter via XFEL

H. Sawada [1] ✉, T. Yabuuchi [2,3], N. Higashi[4], T. Iwasaki[4], K. Kawasaki[4], Y. Maeda[4], T. Izumi[4], Y. Nakagawa[4], K. Shigemori [4], Y. Sakawa[4], C. B. Curry [5,6], M. Frost [5], N. Iwata [4], T. Ogitsu [7], K. Sueda[3], T. Togashi [2,3], S. X. Hu [8], S. H. Glenzer [5], A. J. Kemp[7], Y. Ping[7] & Y. Sentoku[4]

High-intensity, short-pulse lasers are crucial for generating energetic electrons that produce high-energy-density (HED) states in matter, offering potential applications in igniting dense fusion fuels for fast ignition laser fusion. High-density targets heated by these electrons exhibit spatially non-uniform and highly transient conditions, which have been challenging to characterize due to limitations in diagnostics that provide simultaneous high spatial and temporal resolution. Here, we employ an X-ray Free Electron Laser (XFEL) to achieve spatiotemporally resolved measurements at sub-micron and femtosecond scales on a solid-density copper foil heated by laser-driven fast electrons. Our X-ray transmission imaging reveals the formation of a solid-density hot plasma localized to the laser spot size, surrounded by Fermi degenerate, warm dense matter within a picosecond, and the energy relaxation occurring within the hot plasma over tens of picoseconds. These results validate 2D particle-in-cell simulations incorporating atomic processes and provide insights into the energy transfer mechanisms beyond current simulation capabilities. This work significantly advances our understanding of rapid fast electron heating and energy relaxation in solid-density matter, serving as a key stepping stone towards efficient high-density plasma heating and furthering the fields of HED science and inertial fusion energy research using intense, short-pulse lasers.

High-intensity, short-pulse lasers, with intensities exceeding $10^{18}$ W/cm$^2$, efficiently generate high-charge, Mega electron-volt (MeV) electron currents during laser-matter interactions[1,2]. These electron beams not only advance the development of laser-driven electron accelerators[3–5] and the generation of secondary X-ray and γ-ray sources[6,7] but also create matter under extreme conditions. The properties of such materials, resulting from interactions with non-thermal electrons ranging from keV to MeV energies, are of great interest across various disciplines, including astrophysics[8,9], aerospace engineering[10], and both magnetic and inertial confinement fusion (ICF) research[11–13]. Rapidly heated matter by intense short-pulse lasers, with pulse durations less than a picosecond, undergoes an instantaneous increase in electron temperatures while the ions remain cold, leading to a non-equilibrium state[14–16]. This rapid heating process, known as isochoric heating, is pivotal for the fast ignition laser fusion approach[17,18] in ICF[19,20], employing an additional high-intensity laser to generate a beam of energetic charged particles (electrons, protons[21,22], or other ions[23]). These particles deposit their energy directly into a

[1]Department of Physics, University of Nevada, Reno, Reno, NV, USA. [2]Japan Synchrotron Radiation Research Institute, Hyogo, Japan. [3]RIKEN SPring-8 Center, Hyogo, Japan. [4]Institute of Laser Engineering, Osaka University, Suita, Osaka, Japan. [5]SLAC National Accelerator Laboratory, Menlo Park, CA, USA. [6]Department of Electrical and Computer Engineering, University of Alberta, Edmonton, AB, Canada. [7]Lawrence Livermore National Laboratory, Livermore, CA, USA. [8]Laboratory for Laser Energetics, University of Rochester, Rochester, NY, USA. ✉e-mail: hsawada@unr.edu

separately compressed fuel core, initiating fusion reactions for ignition before fuel disassembly. Moreover, matter heated by laser-driven charged particles provides a crucial platform for benchmarking transport and radiative property models, including resistivity[24], opacity[25,26], and ion-stopping powers[27] within the warm dense matter (WDM)[28] regime. This regime lies at the boundaries among classical plasmas, Fermi degenerate matter, and strongly coupled matter. Studying the properties of degenerate matter across different temperatures and densities is crucial for achieving high fusion gains in inertial fusion energy[29], as compressed cold fuels inherently exist in such states[30,31].

Understanding fast electron isochoric heating in solid and high-density plasmas faces long-standing challenges due to the presence of preformed plasma and multiple, competing heating and ionization processes that vary across both space and time. Unlike interactions at non-relativistic intensities[32], a low-intensity pedestal or the rising edge of the main pulse produces a thin plasma layer, preventing the high-intensity pulse from directly interacting with the solid target. Instead, laser-plasma interactions accelerate free electrons in the preplasma predominantly via the J×B force. As illustrated in Fig. 1a, these fast electrons, upon penetrating and moving transversely through a micron-thick solid-density foil, initiate heating and ionization through various mechanisms[33,34]: Joule heating occurs as the fast electrons induce a return current of cooler electrons, leading to resistive heating; drag heating results from direct collisions among non-thermal and background electrons, especially at high densities above $10^{25}$ cm$^{-3}$; and diffusive heating takes place at the interface of the hot preplasma and the cold solid target, primarily near the laser interaction area. Additionally, the mean ionization state (denoted as $\bar{Z}$) is influenced not only by thermal ionization but also by electron impact ionization, meaning that $\bar{Z}$ varies independently from the electron temperature ($T_e$). Consequently, a laser-irradiated thin solid foil exhibits at least three different plasma conditions: a solid-density region near the laser spot primarily heated by Joule and diffusive heating and ionized by thermal and electron impact ionization processes (Region 1), the outside region in the foil affected by Joule heating and impact ionization (Region 2), and the low-density preplasma. Thus, inferring electron

temperatures from space-integrated X-ray emission spectroscopy could be misleading unless the origin of the X-ray sources is identified[35]. To date, indirect measurements and simulations suggest the spatial non-uniformity of dense plasmas within laser-irradiated solid-density foils[36–38]. However, a comprehensive understanding of the dynamics within a solid-density foil under fast electron heating—through the rapid heating and energy relaxation phases between electrons and ions—necessitates direct simultaneous measurements of $T_e$ and $\bar{Z}$ with sufficient spatial and temporal resolutions.

Here, we present the investigation of the spatiotemporal dynamics of laser-driven fast electron heating in a solid-density copper foil, achieving sub-micron and femtosecond resolutions using an X-ray Free Electron Laser (XFEL). The hard X-ray pulses from the XFEL penetrate the preplasma and directly probe the solid-density region of the foil. Leveraging the tunability of XFEL photon energy and the spatial coherence of its beam, we developed a novel X-ray transmission imaging technique to diagnose the solid copper foil heated by an expanding fast electron heat front and the onset of electron-to-ion energy relaxation. Specifically, we inferred the electron temperature of the heated copper foil to be between 7 and 18 eV, based on the smearing observed in an X-ray transmission spectrum near the Cu K-edge. The ionization state, estimated to be between 2.0 and 4.0, was deduced by comparing the time evolution of the heat front with two-dimensional particle-in-cell (PIC) simulations incorporating an electron impact ionization model. Our spatiotemporal measurements and simulations reveal the formation of a solid-density hot plasma the size of the laser spot, surrounded by Fermi degenerate, WDM within a picosecond, and the energy relaxation occurring within the hot plasma over tens of picoseconds. These results not only validate our 2D PIC simulations but also provide important insights into the energy transfer processes in solid-density matter beyond current simulation capabilities. Experimental validation of PIC simulations in fast-electron-heated solid-density targets is a key stepping stone for efficient heating of high-density fusion fuels for fast ignition. This work also underscores the XFEL-based spatiotemporally resolved capability in high-energy-density science using intense short-pulse lasers, paving the way for generating well-characterized WDM and fusion-relevant

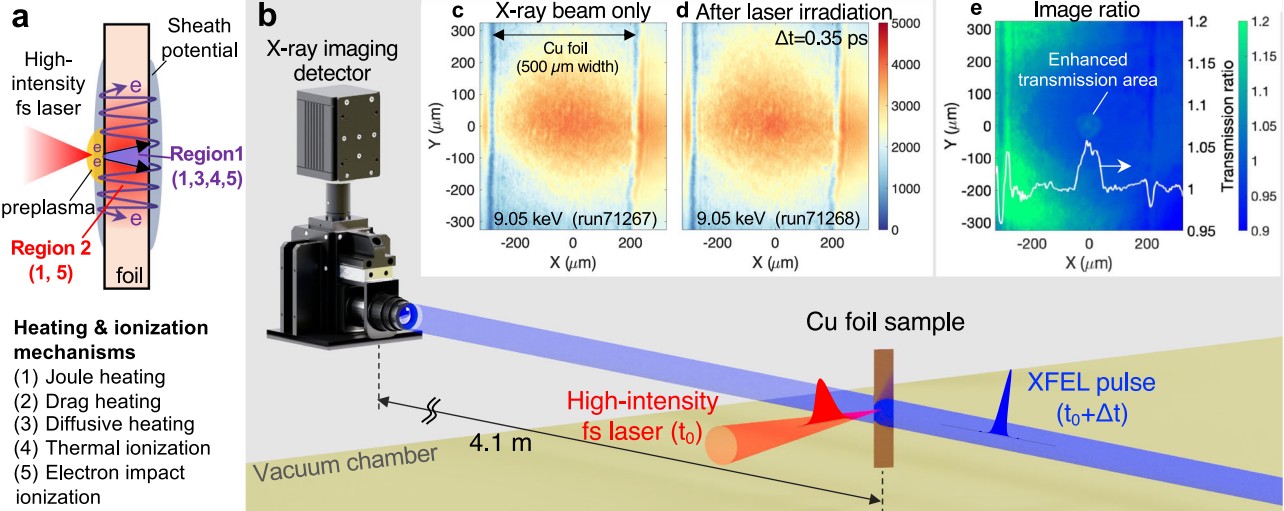

**Fig. 1 | Fast electron isochoric heating mechanisms in a solid foil and experimental layout. a** Schematic illustration of a high-intensity, short-pulse laser interacting with a thin, solid-density foil. Energetic electrons generated in a pre-plasma traverse the foil due to sheath potentials, leading to the creation of two distinct regions attributed to different heating and ionization mechanisms: a high-temperature, solid-density area in Region 1 by Joule and diffusive heating and thermal and impact ionization, and Fermi degenerate warm dense matter in Region

2 by Joule heating and electron impact ionization. **b** Layout of the experimental setup featuring a high-intensity femtosecond (fs) laser and an XFEL pulse. The imaging detector positioned outside the vacuum chamber, ~4.1 m away from the target. Examples of X-ray transmission images recorded (**c**) before and (**d**) after laser irradiation with a delay of 0.35 ps. **e** A ratio of these images provides a raw transmission ratio. An averaged horizontal lineout of the enhanced transmission area is superposed in (**e**).

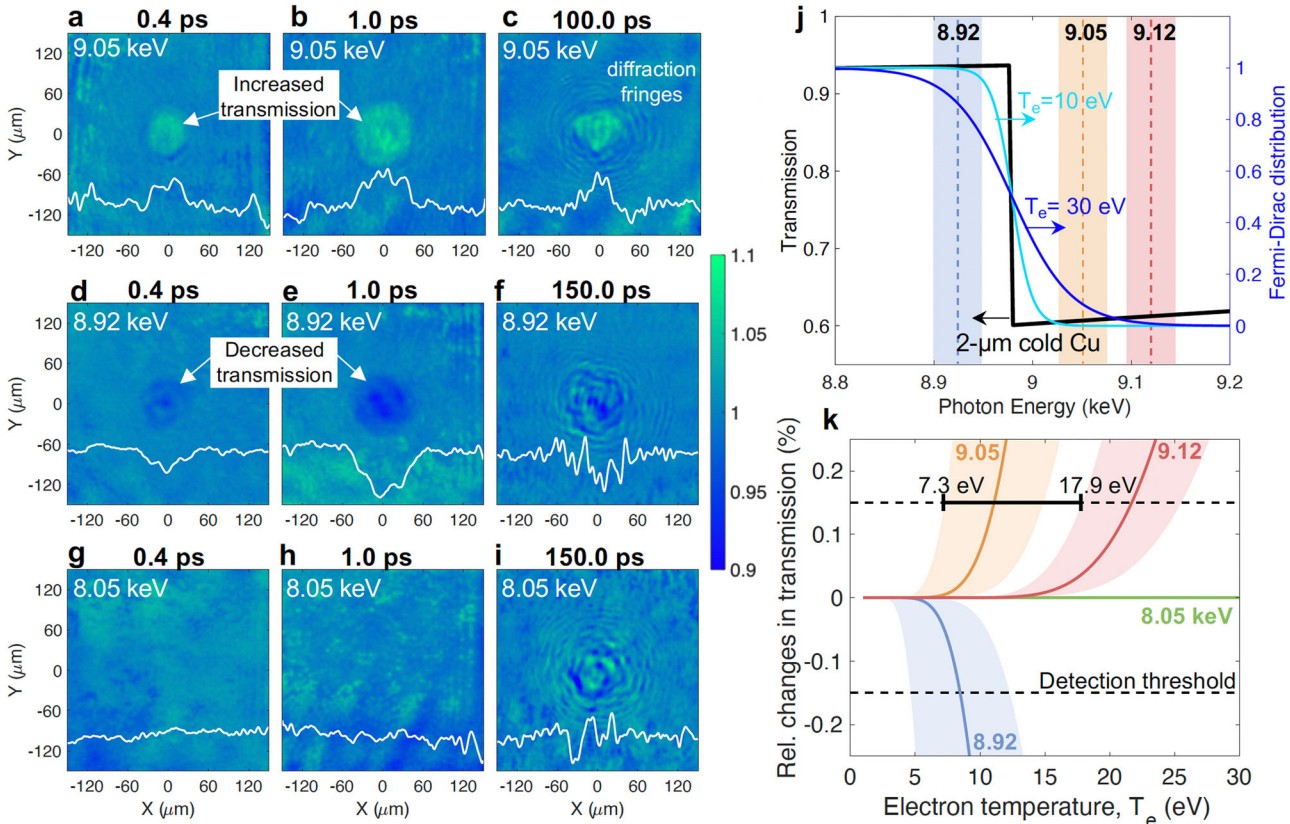

**Fig. 2 | Measured X-ray transmission images and estimating the electron temperature. a–i** Experimental transmission ratio images at 9.05, 8.92, and 8.05 keV for various timing delays. Averaged horizontal lineouts of the image around $Y = 0$ μm are superposed in each image. **j** A transmission profile of a 2 μm thick cold solid Cu foil and the Fermi-Dirac distribution at electron temperatures of 10 and 30 eV. The shaded vertical bars indicate the XFEL's photon energies used with a spectral bandwidth of 50 eV. Transmission changes increase at 9.05 and 9.12 keV, while it decreases at 8.92 keV. **k** Relative changes in transmission are calculated using the Fermi–Dirac distribution as a function of $T_e$. The shading indicates the minimum and maximum temperature range set by the spectral bandwidth for each photon energy.

hot dense matter at temperatures above 1 keV by laser-driven fast electrons.

## Results

### High-intensity femtosecond laser experiment at SACLA

We conducted a pump-probe experiment at Experimental Hutch 6 of the SACLA beamline[39,40] using a high-intensity femtosecond laser and an XFEL beam. As shown in Fig. 1b, the experimental setup consisted of a 2 μm thick, solid-density copper foil irradiated by the laser at ~$2 \times 10^{18}$ W/cm² for fast electron isochoric heating. A collimated X-ray pulse subsequently probed the laser-irradiated foil within a diameter spot of ~1 mm. The resultant 2D X-ray transmission image was captured by a scintillator-based CMOS detector located outside the vacuum chamber. Fig. 1c and d display the X-ray images taken at a probe energy of 9.05 keV before and after laser irradiation, respectively. The raw image (Fig. 1d) exhibits only marginal changes in transmission, which becomes more pronounced in the image obtained by taking the ratio of the two images (Fig. 1e). The non-uniform background was attributed to spatial and intensity fluctuations originating from the self-amplified spontaneous emission source. After correcting for uneven signals, we extracted a horizontal lineout of the transmission-affected area (indicated by the white line in Fig. 1e) to compare with 2D simulations that incorporate the 45° laser incident angle. By varying X-ray probe energies (9.05, 8.92, 9.12, and 8.05 keV) and delay timings from −1 ps to 200 ps, we measured the temporal evolution of the size of the electron-heated area. These transmission ratio images were used to deduce the electron temperature and ionization state within the heated foil. It is noted that the laser irradiation did not alter the solid

copper foil, i.e., by punching a hole or evaporating it, as the X-ray images clearly show the edges of the foil even at a 500 ps timing delay and no diffraction fringes at earlier delay timings. Furthermore, the generation of fast electrons was monitored using a time-integrated X-ray spectrometer, monochromatic X-ray imager, and an electron spectrometer. The mean energy of the fast electron spectrum generated in this experiment was estimated to be ~100 keV, as determined by monochromatic X-ray and XFEL imaging, as reported in our previous publication[41].

### X-ray transmission imaging results

Figure 2 presents a series of transmission ratio images captured at three different X-ray energies (9.05, 8.92, and 8.05 keV), corresponding to timing delays of ~0.4 ps, 1.0 ps, and ~100 ps post-laser irradiation. Notably, the transmission ratio in the area affected by electron heating shows a clear dependence on probe energy: it increases at 9.05 keV (above the K-edge) and decreases at 8.92 keV (below the K-edge), as illustrated by the lineouts. The experimental feature observed at both 9.05 and 8.92 keV expands until ~1 ps, beyond which no significant growth is detected. The analysis of these lineouts and the estimation of width are detailed in the "Method" section of Experimental and Data Analysis Details. The width of the transmission-affected area was found to be consistent at 9.05 and 8.92 keV for the same delay timings. In contrast, at 8.05 keV and 9.12 keV (see below), no change in transmission was observed. The monochromatic Kα imager and X-ray spectrometer verified that the quantity and energy of the fast electrons generated remained consistent across all probe energies. These experimental images were used to infer the electron

temperature and ionization state, a discussion of which follows in subsequent sections. Concentric diffraction fringes, visible in Fig. 2c, f, and i, were observed around 50 ps and at later delays regardless of the probe energies. These fringes likely result from the spatially coherent X-ray beam illuminating a locally ablated target surface. Qualitative comparisons between the measured and calculated fringe patterns suggest that the size of the ablated region is on the order of the laser spot (~26 μm in diameter). This finding, much smaller than the measured transmission-affected area of ~120 μm presented later, implies that only the region near the laser spot is significantly heated, radiatively cooled, and thermalized, causing the ablation over tens of picoseconds, while the ions in the outside region remain within the initial foil (see the "Diffraction fringe analysis" of the "Methods" section). The ratio of ~5 between the size of the hot plasma region and the transmission-affected area is consistent with past literature[36]. It is important to note that the 50 ps threshold represents an upper bound for the electron-ion energy relaxation and may be specific to this particular laser-target interaction. Further insights into the size of the deformed area, along with the ion temperature and collision frequency in the intensely heated region, are detailed in the "Methods" section (collision frequency in WDM).

### Electron temperature estimation

The transmission ratio images in Fig. 2, varying with probe energy, enable the determination of the electron temperature in the heated area. This process leverages the sensitivity of the K-edge transmission (absorption) spectrum to electron temperature in Fermi degenerate materials like metals—a phenomenon explored in previous research[42–45]. Theoretically, the transmission slope near the K-edge is predicted to follow a Fermi–Dirac distribution, expressed as $f(E) = \left[1 + e^{(E-\mu)/kT_e}\right]^{-1}$, where $\mu$, the chemical potential, is approximately equal to the Fermi energy ($E_F$). Figure 2j shows the X-ray transmission profile for a 2 μm thick, cold solid-density Cu foil alongside the Fermi-Dirac distribution for 10 and 30 eV electron temperatures. Our measurements of the changes in transmission at 8.92 and 9.05 keV reflect the changes in electron temperature of the distribution, underscoring that the target remains in a degenerate state. The absence of transmission change at 9.12 keV suggests an upper-temperature limit. To quantitatively determine a range of electron temperatures, we calculated the relative changes in transmission at each X-ray probe energy by varying electron temperature, as illustrated in Fig. 2k. Given a detection threshold at 5% of the peak signal, we determined that our imaging technique can detect relative signal changes above 0.15%—details are provided in the "Experimental and data analysis details" of the "Method" section. This analysis allowed us to deduce a lower temperature limit of 7.3 eV, necessary to observe transmission changes at both 8.92 and 9.05 keV, and to identify an upper temperature bound of 17.9 eV, the minimum temperature at which transmission changes at 9.12 keV become discernible. This marks the first determination of electron temperature bounds from the slope of the smeared K-edge profile in fast electron heated targets.

### 2D particle-in-cell simulations

To understand the dynamics of fast electron heating mechanisms and determine the ionization state of the electron-heated area, we performed two-dimensional collisional PIC simulations using the PICLS code[46,47]. Figures 3a and b illustrate these simulations, providing time-dependent spatial profiles of electron temperature ($T_e$) and ionization state ($\bar{Z}$), averaged over a 2 μm thick Cu target, with further details available in the "Methods" section (2D PIC Simulation Details). The evolution of the $T_e$ and $\bar{Z}$ profiles proceeds as follows: immediately after the laser irradiation onto the foil, which generates fast electrons, Joule heating occurs near the focal spot (around $Y=$ ~0 μm) as these electrons penetrate the depth of the target. Upon reaching the rear of the target, most electrons are confined by the sheath potential and

traverse laterally, raising the temperature to a range of 6–10 eV. This simulated peripheral temperature is consistent with the experimentally deduced temperature presented in Fig. 2k. Despite multiple collisions between fast and bulk electrons due to electron recirculation in the thin foil, the temperature outside the intensely heated region remains unchanged, indicating a negligible impact from drag heating in solid-density matter. After 0.56 ps, an increase in electron temperature near the laser interaction region suggests that diffusive heating is occurring, caused by the thermal gradient between the colder solid and the warmer preplasma. The diffusion process begins after the laser pulse has ended because laser-induced magnetic fields trap electrons within the preplasma. Additionally, Fig. 3a and b reveal a noticeable discrepancy between the spatial distributions of $T_e$ and $\bar{Z}$, attributed to electron impact ionization.

### Ionization state estimation

As the simulations suggest, the observed changes in transmission can be more accurately explained by the evolution of the ionization profile, which broadens over time, rather than by the temperature profiles. Figure 3c illustrates the progression of the transmission-affected area within the copper foil, as recorded at 9.05 keV and 8.92 keV, alongside simulated ionization states. During heating, the expansion of the heated area approaches the speed of light, indicating that the heat front is driven by near-relativistic electrons[41]. This rapid expansion stagnates after ~0.5 ps, and the experimental feature remains unchanged until the appearance of diffraction fringes around 50 ps (with further details up to 100 ps provided in the "Experimental and data analysis details" in "Methods"). In Fig. 3c, the experimental measurements are bounded by the simulations with an ionization state ($\bar{Z}$) ranging from 2.0 to 4.0. Note that the inferred ionization state occurs behind the propagating heat front, with the overall spatial ionization profile peaking in the high-temperature region, as shown in Fig.3b. The relatively large measurement uncertainties in width, particularly noticeable during the heating phase (e.g., at ~0.3 ps), may be due to laser arrival time drift over several hours[40].

## Discussion

Our study, utilizing spatiotemporally resolved measurements alongside 2D PIC simulations, has unveiled the rapid transitions of solid-density cold copper foils to highly charged hot plasma and WDM. Figure 4 illustrates the temporal evolution of the foil's conditions on an electron temperature-density diagram, categorized by the electron coupling parameter, $\Gamma_e$, and the degree of degeneracy, $\Theta$, which are defined as $\Gamma_e = (4\pi n_e/3)^{1/3} e^2/k_B T_e$ and $\Theta = k_B T_e/k_B T_F$, where $k_B$ is the Boltzmann constant and the Fermi temperature, $T_F$, is $(\hbar^2/2m_e k_B)(3\pi^2 n_e)^{2/3}$. Our measurements show the formation of a highly ionized hot region near the laser focal spot, surrounded by Fermi degenerate, WDM. The transition to these conditions is described by the trajectories derived from the simulated spatial profiles at $Y = 0$ μm and $Y = 60$ μm in Fig. 3a and b, as representatives for Region 1 and Region 2. Near the laser focal area ($Y = 0$ μm), the foil undergoes rapid heating to high temperatures through Joule heating (indicated by the purple arrow), evolving into diffusive heating above 100 eV. This region experiences changes in ionization state due to both thermal and electron impact ionization, consequently increasing the electron density through $n_e = n_i \cdot \bar{Z}$ even though the ion density remains constant. The plasma condition in Region 1 is non-degenerate ($T_e > T_F$), so the smearing of the X-ray spectrum near the K-edge was not applicable. In contrast, the peripheral region at $Y = 60$ μm is subject to rapid Joule heating to several eV in ~0.1 ps, followed by electron-impact ionization (shown by the red arrow), reaching up to $\bar{Z} \sim 3$, with a slight subsequent temperature increase. Our measurements of this peripheral region agree with the simulation results, as indicated by a white box in the figure. The simulations indicate that measuring the initial rise in the electron temperature due to Joule heating requires a faster resolution

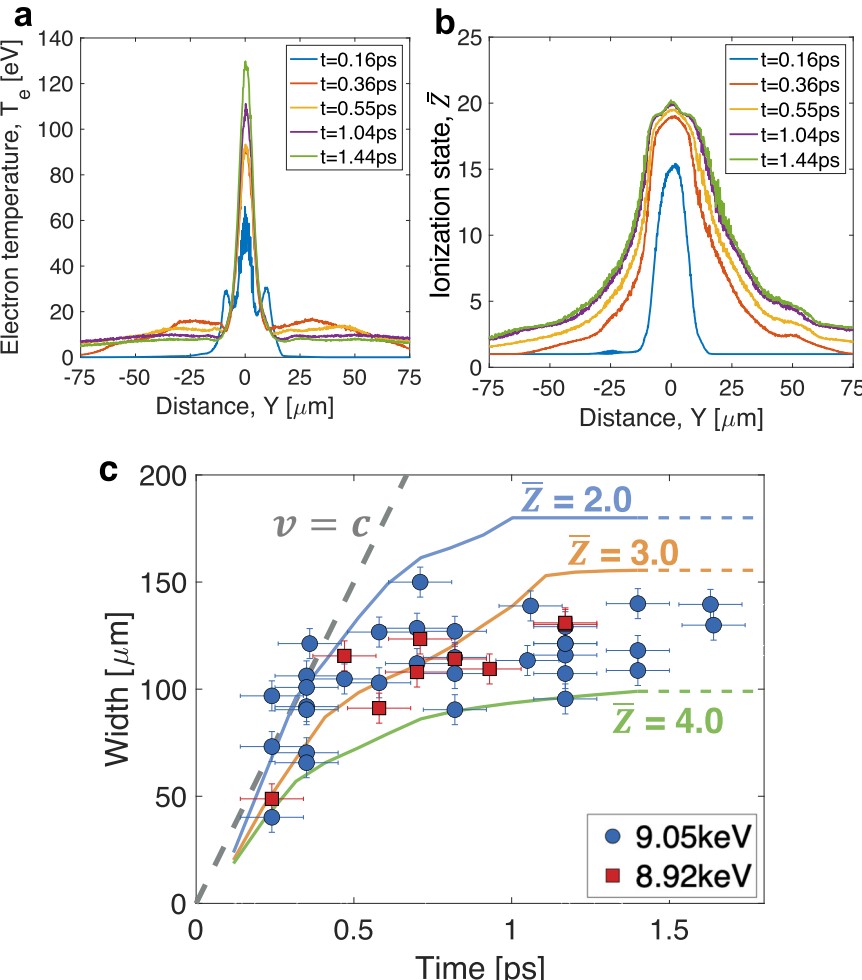

**Fig. 3 | Results of 2-D particle-in-cell simulations. a** Simulated electron temperature and (**b**) average ionization state profiles averaged over a 2 μm thick solid Cu foil from 2D PIC simulations (see Fig. 9). The evolution of these profiles is presented from 0.16 ps to 1.44 ps. Deviations of the ionization state profile from the temperature profile indicate the contribution of electron impact ionization. **c** Comparison of the widths of the measured transmission-affected area with simulated ionization states. The circle and square symbols represent the data at 9.05 keV and 8.92 keV, respectively. A dashed line indicates the speed of light (v = c) for reference. The error bars in time are ± 0.1 ps based on the uncertainty of the XFEL and the optical laser timing jitter, while the errors in width are ± 7.9 μm due to averaging of the experimental line profiles. The solid lines are results from the PIC simulation, and a constant extension of the simulation is assumed as extended dashed lines after ~1.4 ps.

than our current 0.1 ps. The target conditions of the other parts of the foil fall between these two extremes. This comparison validates the Joule heating and electron impact ionization physics models incorporated in our numerical code. However, further experiments are needed to verify the high-temperature plasma and to examine the effects of spatial gradients in the depth direction of the foil.

The temperature-density diagram visualizes the two distinctly different high-energy-density states accessible with an intense short-pulse laser for unique applications. Diffusive heating plays a pivotal role in generating high-temperature, highly charged plasmas. In this experiment, using a 0.64 J, 40 fs optical laser, we achieved an efficiency of ~0.6% in heating the solid-density copper foil within a 4 μm spot diameter above 100 eV. Further research with state-of-the-art peta-watt-class, short-pulse lasers could facilitate the creation of hot dense matter exceeding temperatures of 1 keV[48], providing crucial insights into the underlying physics for fast ignition laser fusion. Additionally, Fermi-degenerate, WDM, produced by significant electron impact ionization, exhibits continuously evolving ionization states in the transverse direction of the foil, while the electron temperature remains nearly constant as shown in Figs. 3a and 4. These regions, where ionization states are the primary variable, could serve as an ideal testbed for benchmarking quantum molecular dynamics and atomic

physics calculations, including models of ionization potential depression[49] for dense matter.

This work has demonstrated the XFEL's ability to perform detailed, spatiotemporally resolved measurements at the sub-micron scale and femtosecond resolutions in high-intensity short-pulse laser experiments. These diagnostic capabilities, vastly more precise than those typically employed in high-energy-density and ICF experiments[29], play a crucial role in closing existing knowledge gaps. Importantly, the versatility of this measurement technique across various FEL beam wavelengths paves the way for future investigations with next-generation XFEL facilities, such as MEC-U[50].

## Methods
### Experimental and data analysis details
In experimental hutch 6 of the SACLA XFEL facility, the high-intensity femtosecond laser system comprises two beams, each with a potential of 0.5 PW. One beam, delivering 12.5 J in 25 fs, has successfully commissioned and operates in conjunction with the XFEL beam, producing 8 J in 40 fs. However, due to damage in the laser optics, the energy delivered to the target in this experiment was limited to 0.6 ± 0.1 J. The pulse duration and laser spot size were 40 fs (full width at half maximum, FWHM) and 15 μm × 20 μm (FWHM), respectively. This resulted

in a peak intensity on the target of $I = 2 \times 10^{18}$ W/cm$^2$ and a relativistic factor ($\gamma$) of 1.2, where $\gamma$ is defined as $\sqrt{1 + a_0^2/2}$ and $a_0$ is $0.85\sqrt{I\lambda_{\mu m}^2/10^{18}}$.

For the pump-probe experiment, the incident angles for the high-intensity optical pump laser and the XFEL probe laser were set at 45° and 27°, respectively, relative to the normal of a 2 μm-thick, $0.5 \times 3.0$ mm$^2$ copper strip. The X-ray pulse had a nominal duration of ~10 fs and an energy of 0.6 mJ. The timing delay of the XFEL beam with respect to the fs laser varied from −0.1 ps to 200 ps. Additionally, X-ray transmission images of the laser-irradiated copper strips were recorded at the photon energies of 8.92, 9.05, 8.05, and 9.12 keV with a spectral bandwidth of ~50 eV. Figure 5 shows the transmission ratio images with the horizontal lineouts at 9.12 keV for delays of 0.4, 1.0, and 100 ps. At earlier delays, no clear variations in transmission were

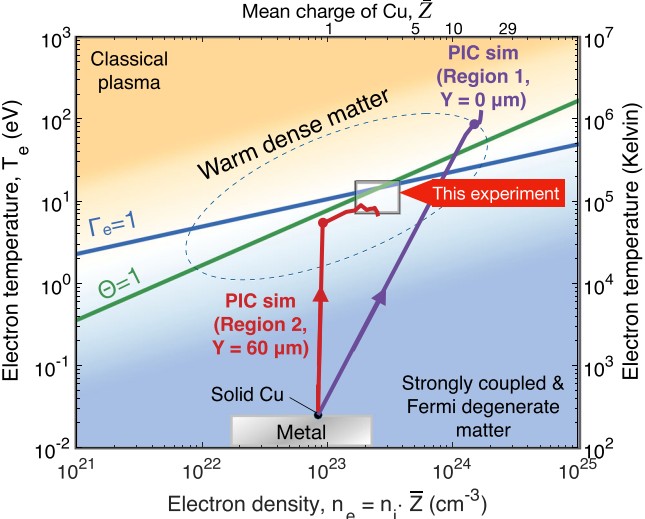

**Fig. 4 | Electron temperature and density contour for isochorically heated matter using an intense short-pulse laser.** The temperature and density map is divided using the coupling constant, $\Gamma_e$, and the degree of degeneracy, $\Theta$, both of which are defined in the text. The shading indicates that matter at high temperatures and low densities with $\Gamma_e \ll 1$ and $\Theta > 1$ is treated as classical plasmas, while cold, dense matter ($\Gamma_e > 1$ and $\Theta < 1$) is considered to be strongly coupled and Fermi degenerate matter. The warm dense matter regime is loosely defined in the vicinity of $\Gamma_e \sim 1$ and $\Theta \sim 1$. The experimentally deduced conditions are depicted with a white box. Trajectories represented by red and purple lines correspond to PIC simulations at different distances from the laser interaction region, aligning with the axes in Fig. 3a and b. The evolving conditions at $Y = 0$ μm and $Y = 60$ μm are presented as representatives of conditions near the laser interaction in Region 1 and further from the center in Region 2. Filled circles on the solid lines indicate the simulation time of 0.15 ps.

observed, whereas concentric fringe patterns appeared after ~50 ps, similar to the case at 8.05 keV. The shot rate for the experimental campaign was approximately one shot every 3 min.

Time-integrated 2D monochromatic Kα images and X-ray spectral measurements were used as follows: The mean energy of the fast electrons (~100 keV) was estimated based on the radius of the Kα emission spot compared to the electron's collisional stopping range, calculated using the continuous-slowing-down approximation[51]. With the X-ray spectrometer, we recorded characteristic emission lines at 8.05 keV (Cu Kα) and 8.9 keV (Cu Kβ). However, no line emissions from highly ionized copper ions, such as Heα, Lyα, or ionized Kα lines, were observed. This information indicates an upper bound of the electron temperature within the copper foil.

The lineouts of the transmission ratio image, as presented in Fig. 2a and d, are further detailed in Fig. 6 to illustrate how the width of the transmission-affected area was estimated. After correcting for the non-uniform background and setting the base signal to be zero, we first calculated a threshold level at 5% of the averaged peak signals[41]. This detection threshold corresponds to thresholds in transmission changes after the base adjustment to be between 0.15% and 0.23% in this experiment. With these thresholds, we determined not only the width of the experimental signature as shown in Fig. 6a and b but also the minimum thickness variation detectable by our imaging technique. Transmission through a 2 μm thick solid-density copper foil at 8.05 keV is 91.3%. Using a tabulated absorption coefficient[52], we calculated relative changes in X-ray transmission, as illustrated in Fig. 6c, indicating that our measurement technique is sensitive to foil thickness changes greater than 32.4 nm. This minimum thickness threshold was later used to discuss the collision frequency and ion temperature.

Figure 7 presents an extended comparison of the width of the experimental features with the simulated ionization states over a period of up to 100 ps from Fig. 3c. The experimental widths at 1 ps and 10−20 ps are similar, suggesting that no significant target disassembly occurs before 10 ps. This observation is supported by the appearance of concentric diffraction fringes around 50 ps. Data that are strongly interfered with diffraction fringes for delays between 50 and 200 ps are excluded. The notable shot-to-shot variations observed, particularly at 10 and 100 ps, are likely due to fluctuations in laser conditions and target positions. The laser arrival time is known to drift over several hours, even within a single shot cycle[40], which also likely contributes to the data scatter observed at 0.2−0.4 ps. Additionally, these variations could occur when measurements are taken in different shot cycles, such as one measurement during a single shot cycle and another following the venting of the vacuum chamber and adjustment of the laser focus spot. The experimental data, particularly after 1.5 ps, are bounded by ionization states between $\bar{Z} = 3.0$ and $\bar{Z} = 4.0$. However, extrapolating the simulated ionization state as constant over 100 ps oversimplifies the ionization process. Further details such as

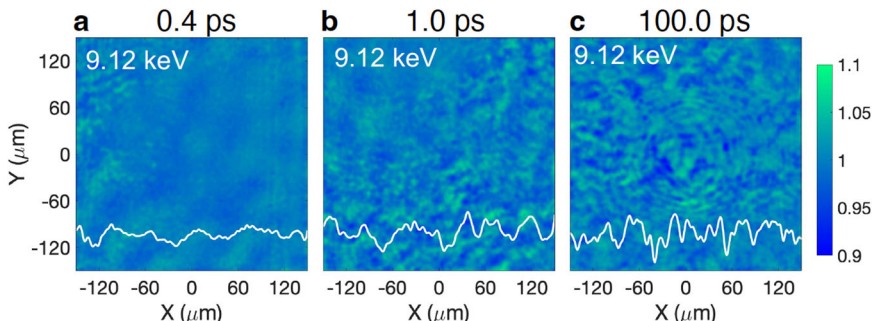

**Fig. 5 | Measured X-tray transmission ratio images at 9.12 keV.** The transmission ratio images are recorded using a photon energy of 9.12 keV at timing delays of (**a**) 0.4 ps, (**b**) 1.0 ps, and (**c**) 100 ps.

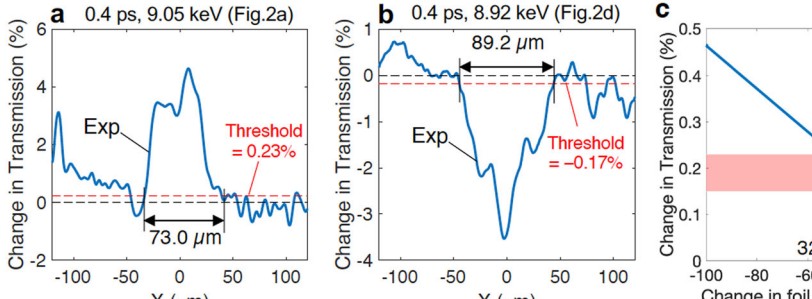

**Fig. 6 | Analysis of the experimental lineouts and detection threshold.** The lineouts of the experimental features shown in Fig. 2a and d for 9.05 keV and 8.92 keV are presented in (**a**) and (**b**). The width of the transmission-affected region is estimated using a threshold set at 5% of the peak signal. Due to fluctuations in the measured signals, the detection threshold varied between 0.15% and 0.23% in transmission changes over the data analyzed. **c** An estimate of the minimum

changes in foil thickness is considered by calculating changes in X-ray transmission at 8.05 keV through a solid copper foil with a various thickness of (2 μm − ΔL). The shading indicates the detection threshold range (0.15–0.23%). With these thresholds, this figure suggests that our imaging technique detects changes in transmission when the foil thickness varies from the initial 2 μm thickness by more than 32.4 nm.

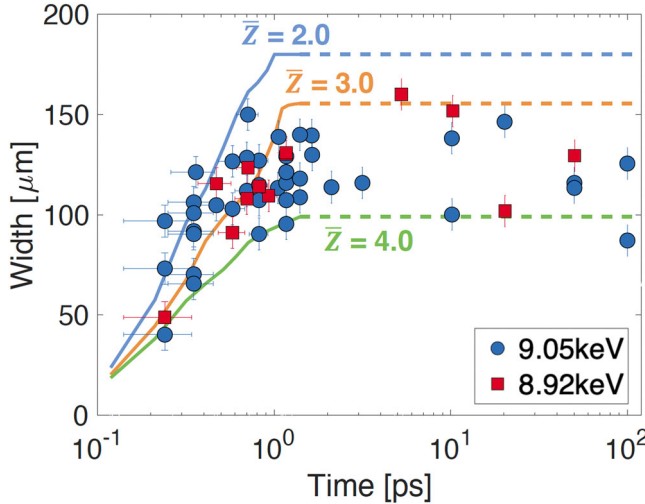

**Fig. 7 | Comparison of the extended experimental data upto 100 ps.** The time history of the widths of the measured transmission-affected (ionized) regions is compared with simulated ionization states ranging between $\bar{Z} = 2.0$ and 4.0. The circle and square symbols correspond to the data at 9.05 keV and 8.92 keV, respectively. The error bars in time and in width are ± 0.1 ps and ± 7.9 μm, respectively. Solid lines represent the PIC simulations up to 1.4 ps. Beyond this point, the simulation results are assumed to remain constant, as indicated by dashed lines.

radiative cooling, recombination, and cooling by plasma expansion need to be considered.

### Fresnel diffraction calculations

Concentric fringe patterns consistently appear at delays exceeding 50 ps, as shown in Fig. 2c, f, and i. These fringes are likely caused by the 2D coherent X-ray pulse illuminating a planar foil, with the deformation of the target surface resulting from ion motion (ablation), rather than from surface plasmons or laser-induced modulations. This is supported by Fig. 2f, where at least two sets of circular fringe patterns are overlapped at 150 ps. Here, we present Fresnel diffraction calculation results to estimate the approximate size of the ablated region within a 2 μm thick solid-density copper foil by qualitatively comparing the measured and calculated diffraction patterns. The experimental setup is essentially equivalent to a classical diffraction experiment, wherein a monochromatic, coherent plane wave strikes a semi-transparent flat target featuring a locally ablated region, differing from the scenario with an opaque aperture. Given the detector distance (*L*) of 4.12 ± 0.01 m from the target and the light wavelength (λ)

of 0.154 nm (corresponding to 8.05 keV), the Fresnel number ($N_F$), defined as $\pi a^2/(L \cdot \lambda)$, was calculated to range from 0.12 and 12.38 for a characteristic aperture size (radius, *a*) ranging from 5 to 50 μm. Therefore, Fresnel diffraction calculations are applicable in this near-field diffraction regime, where $N_F$ is greater than approximately 1[53].

We modeled two-dimensional Fresnel diffraction of a circular transparent aperture using the transfer function method[54]. This approach was specifically utilized to compare with the image captured at 8.05 keV with a 50 ps timing delay, as shown in Fig. 8a. In our model, a plane wave with a wavelength of 0.154 nm illuminates a two-dimensional plane containing a circular spot. This spot has a transmission slightly different from the cold transmission of a 2 μm solid Cu foil at 8.05 keV. The model calculations for aperture radii of *a* = 5, 13, and 25 μm, alongside comparisons with the experimental measurements, are displayed in Fig. 8b–e. These comparisons reveal that the position of the first bright ring, as indicated by arrows in the figures, shifts outward with increasing aperture size, consequently affecting the diffraction intensity patterns. The calculations with an aperture radius of 13 ± 2 μm showed a qualitative agreement with the experimental measurement, as marked by the vertical dashed lines in Fig. 8e. The inferred size of the ablated region ( ~ 26 μm in diameter) corresponds closely to the laser spot size and a high-temperature region (near *Y* = 0 μm or Region 1), as illustrated in Figs. 3a and 1a. This observation provides direct evidence of which part of the heated target begins to disassemble first following equilibration.

The timing of the electron-ion equilibration from our measurements is consistent with findings from other studies that have established an upper limit for the surface ablation timing[55,56]. In particular, our measurements are close to the duration of time-resolved Kα emission measurements (10–20 ps)[57] rather than that of K-shell emissions (several ps)[36,37,58], indicating that the hot plasma that produces K-shell emissions (Heα, Heβ, Lyα, etc) during a high-intensity, laser-solid interaction is radiatively cooled, which diminishes the X-ray emission, before target expansion occurs. In the next section, we calculate the evolution of electron temperature and ionization state due to radiative cooling and consider it in estimating an electron-ion collision frequency in warm dense copper.

### 2D PIC simulation details

We performed two-dimensional PIC simulations using the fully relativistic, collisional PIC code PICLS, as depicted in Fig. 3. The simulation setup and contours of electron temperature and mean ionization state at 0.16 ps is shown in Fig. 9. PICLS self-consistently solves the relativistic equations of motion for numerous particles and Maxwell's equations, addressing laser-target interactions, high-energy electron generation, and electron transport. It includes modules for Coulomb

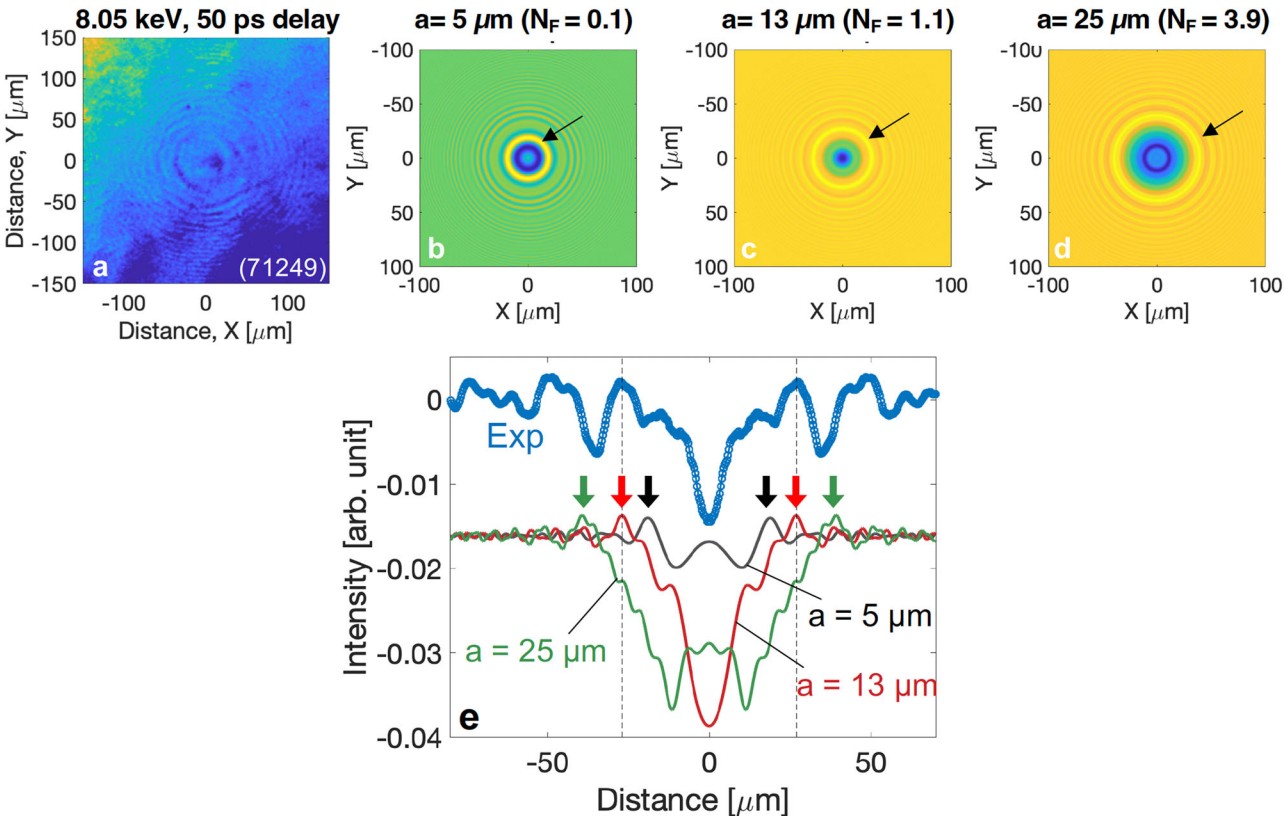

**Fig. 8 | Results of Fresnel diffraction calculations. a** A measured transmission ratio image at 8.05 keV at a timing delay of 50 ps. Fresnel diffraction intensity patterns are calculated with various aperture sizes, specifically aperture radii of (**b**) 5 μm, (**c**) 13 μm, and (**d**) 25 μm. **e** Comparison of the azimuthally averaged measured fringe profile with the calculations. The arrows indicate the spatial positions of the first bright fringe for the different aperture radii.

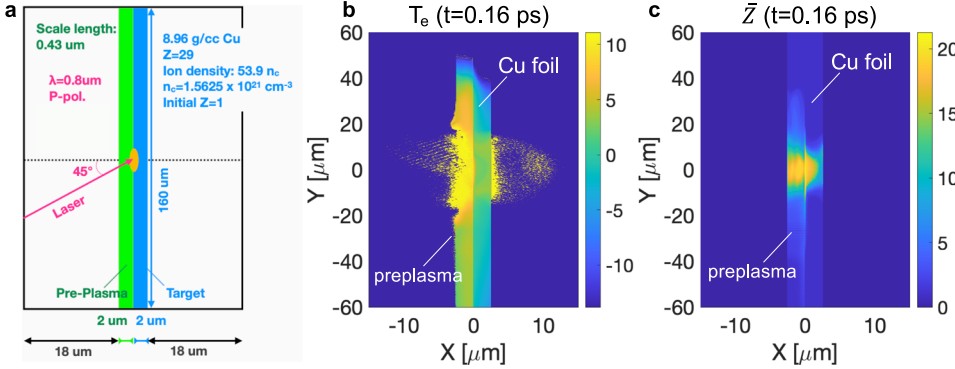

**Fig. 9 | PIC simulation setup. a** A schematic of the 2D PIC simulation setup. Contours of (**b**) the simulated electron temperature on a log scale and (**c**) the ionization state at 0.16 ps.

collisions and dynamic ionizations, the primary ionization mechanisms. The simulations used the experimental laser conditions: an 800 nm wavelength, a 40 fs Gaussian pulse (FWHM), P-polarized light at a 45° incidence angle, and a peak intensity of $2 \times 10^{18}$ W/cm². The simulation includes a preplasma with a scale length of 0.43 μm within a 2 μm layer. The simulation encompassed a $40 \times 160$ μm² system box with $2000 \times 8000$ cells, running up to -1.5 ps. The simulation predicts that the high-temperature solid-density plasma in Region 1 reaches $T_e = 120$ eV and $\bar{Z} = 20$. The collision frequency and electron-to-ion energy relaxation time are calculated to be $4.9 \times 10^{16}$ s⁻¹ and 1–2 ps, respectively. The latter value is an order of magnitude smaller than the observation 50 ps in this experiment. This discrepancy provides

further evidence that radiative cooling must occur prior to energy relaxation.

**Radiative cooling calculations**

The evolution of the target conditions due to radiative cooling is calculated over tens of picoseconds, beyond the capabilities of PIC simulations. We computed the radiative cooling of non-degenerate plasmas over time by numerically solving the equation $n_e(t)\frac{dT_e(t)}{dt} = \eta(T_e(t), n_i)$, where $\eta(T_e(t), n_i)$ represents the emissivity. This emissivity, accounting for free-free, bound-free, and bound-bound emissions across various electron temperatures at the solid copper ion density, was precomputed using the collisional-radiative

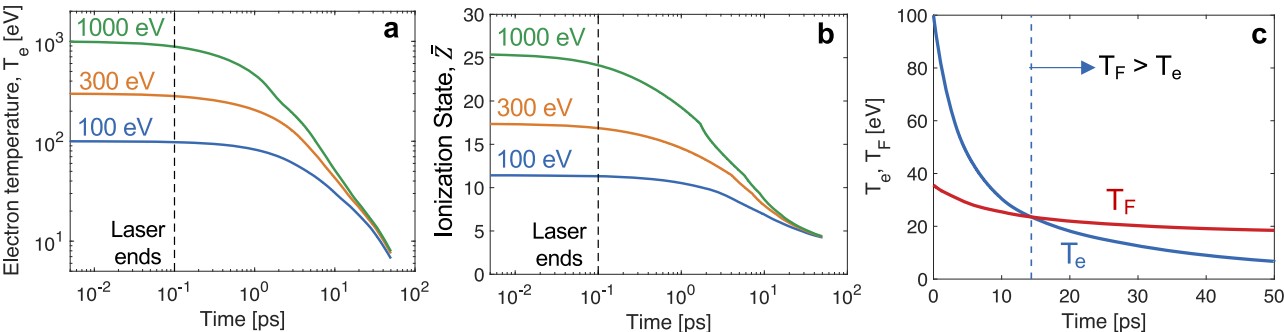

**Fig. 10 | Radiative cooling calculations.** Radiative energy loss of a solid-density hot Cu plasma is calculated for initial electron temperatures of 100, 300, and 1000 eV up to 50 ps. The temporal evolution of (**a**) the electron temperatures and (**b**) the ionization states shows significant decreases in both quantities between 1 and 10 ps. **c** Comparison of the temporal decay in the electron temperature and Fermi temperature with constant density. The rapid drop in the electron temperature leads to the transition of the plasma to a degenerate state ($T_F > T_e$). The vertical dashed line indicates the transition time of 14.3 ps for the 100 eV case.

atomic physics code FLYCHK[59] for temperatures between 100 eV and 1 keV. The electron density, $n_e(t) = \bar{Z}(T_e) \cdot n_i$, was updated based on FLYCHK results. Figure 10 illustrates the time evolution of $T_e$ and $\bar{Z}$ for plasmas with initial electron temperatures of 100, 300, and 1000 eV. Notably, $T_e$ significantly decreases within several picoseconds after the Gaussian laser pulse ends around 0.1 ps, as indicated by a dashed vertical line.

In high-temperature, thermally ionized plasma (e.g., around $Y = 0$ μm in Fig. 3a), the ionization state decreases over time with $T_e$. For example, in the 100 eV case, $T_e$ falls below the Fermi temperature ($T_F$) after 14.3 ps, as shown in Fig. 10c, because the decrease in $T_e$ is faster than that in the ionization state. The corresponding transition times from non-degenerate to degenerate plasmas for 300 eV and 1000 eV are 18.4 ps and 20.1 ps, respectively. These results suggest that even a highly charged, thermally ionized hot plasma, typically considered classical, can transition into Fermi degenerate matter due to radiative cooling over tens of picoseconds, irrespective of the initial temperature. Considering multi-dimensional effects, this transition could occur even more rapidly. Thus, it is plausible that the target near the laser interaction region ($|Y| < \sim 13$ μm in Fig. 3a) was in a degenerate state when the surface ablation occured. The radiative cooling calculations show that a plasma with an initial condition of $T_e$ of 100 eV and $\bar{Z}$ of 12 decreases to $T_e$ of 6.7 eV and $\bar{Z}$ of 4.3 at 50 ps. These values at 50 ps are used in the following section to infer the ion temperature and an empirical parameter for the electron-ion collision frequency in Fermi degenerate copper plasma.

### Collision frequency in warm dense matter
The electron-ion collision frequency for WDM ($\nu_{wdm}$)[60,61] significantly differs from that for ideal plasmas and is given by

$$\nu_{wdm} = K_{wdm} \frac{e^2}{\hbar \nu_F} \frac{k_B T_i}{\hbar} \tag{1}$$

where $T_i$ is the ion temperature, $k_B$ is the Boltzmann constant, $\nu_F$ is the Fermi velocity, and $K_{wdm}$ is an empirical constant. In this section, we determined the ion temperature from the velocity of ablated ions and an empirical constant, $K_{wdm}$, in the above equation.

The characteristic time, $\tau$, of ~50 ps when the significant ion motion is observed allows us to estimate the collision frequency at $1.2 \times 10^{15}$ s$^{-1}$ through $\tau = M_i / 2m_e \nu_{ei}$, where $M_i$ and $m_e$ are the ion and electron masses, respectively. The Fermi velocity, defined as $\nu_F = (3\pi^2 n_e)^{1/3} \hbar / m_e$, changes in proportion to the ionization state at a constant density. In the previous section, the radiatively cooled plasma condition shows a $T_e$ of 6.7 eV and $\bar{Z}$ of 4.3 at 50 ps. Using this ionization state and the initial ion density, we calculate the Fermi velocity to be $2.6 \times 10^8$ cm/s.

Our imaging technique, as shown in Fig. 6c, detects changes in thickness once the 2 μm thick foil thickness changes more than 32.4 nm. We hypothesize that all ions within a 32.4 nm depth (and a 13 μm radius) are ablated around 50 ps, leading to an ion velocity of $6.5 \times 10^4$ cm/s and a corresponding ion temperature of 0.28 eV. Applying these values to Eq. (1), we found the $K_{wdm}$ constant to be ~3.2. Several values of the empirical constant have been reported: 4.6 for solid-density, room-temperature aluminum, and 18.8 for warm dense aluminum under equilibrium plasma conditions ($T_e = T_i$)[60]. Although further research is required to accurately determine the electron-ion collision frequency in warm dense copper, this work has demonstrated that the XFEL-based time-resolved measurements provide new insights into electron-ion energy relaxation processes and subsequent hydrodynamic expansion induced by high-intensity, femtosecond lasers at both non-relativistic and relativistic intensities[62].

### Data availability
Experimental data were generated at the SACLA XFEL facility. The key data underpinning the conclusions of this work are included in the article. All raw data relevant to the findings of this study are available from the corresponding author (H.S.) upon request.

### Code availability
The authors declare that the computer code PICLS supporting the findings of this study is fully documented within the paper and its references. Additional inquiries about the codes should be directed to Y. Sentoku.

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

## Acknowledgements

The XFEL experiments were performed at the BL2 of SACLA with the approval of the Japan Synchrotron Radiation Research Institute (JASRI) (Proposal No. 2019A8002). This material is based on the work supported by the National Science Foundation under Grant No. 1707357 and 2010502 through the NSF/DOE Partnership in Basic Plasma Science and Engineering. Y. Sentoku and N.I. are supported by JSPS Kakenhi Grants No. JP20H00140, 20K14439, 24H00204, and JP19KK0072. T.Iwasaki, K.K., Y.M., and K.Shigemori are supported by JSPS Kakenhi Grants No.17H02996. Part of the work was performed under the auspices of the US Department of Energy by Lawrence Livermore National Laboratory under contract no. DE-AC52-07NA27344 (A.J.K., T.O., and Y.P.) and funded by the DOE Office of Science Early Career Program under SCW 1265 (A.J.K.) and SCW 1420 (Y.P.). The work of C.B.C., M.F., and S.H.G. was supported by the U.S. Department of Energy (DOE), Office of Science, Fusion Energy Sciences under FWP 100182. C.B.C. was also partially supported by the Natural Sciences and Engineering Research Council of Canada (NSERC). T. Izumi, Y. N., and Y. Sakawa were supported by JSPS Kakenhi Grants No.17H06202 and JSPS Core-to-Core Program B: Asia-Africa Science Plat-forms Grant No. JPJSCCB20190003. S.X.H acknowledges the support of the Department of Energy under award number DE-NA0004144. T.Y. is supported by JSPS Kakenhi Grants No. 19K03788 and 22K03571.

## Author contributions

H.S. led the writing the manuscript with the help of the coauthors' inputs and feedback. H.S., T.Y., T.Iwasaki, K.K., Y.M., T.Izumi, Y.N., K.Shigemori, and Y.Sakawa, performed the experiment and acquired the data. N.H., N.I., Y.Sentoku, S. X. H., and H.S. performed the theoretical and numerical work. T.Y., K.Sueda, and T.T. operated the high-power femtosecond laser. C.B.C and M.F. performed laser cutting of the Cu foils. H.S., S.H.G., A.J.K., Y.P., T.O., and Y.Sentoku developed the original experimental concept. All authors contributed to the work presented here and to the final manuscript.

## Competing interests

The authors declare no competing interests.
