## [Peer Review File · Nature Communications]

Spatiotemporal Dynamics of Fast Electron Heating in Solid Density Matter via XFELReviewers' comments:

Reviewer #1 (Remarks to the Author):

In this manuscript, the authors describe a series of experiments conducted with a short-pulse (fs scale) high-intensity ($2 \times 10^{18} \text{ W cm}^{-2}$) laser at SLAC. The experiments involved small copper targets that radiate characteristic emission near 9 keV where the K-edge is. The novelty in this work arises from the use of the XFEL beam available at SLAC as a diagnostic to provide high spatial and temporal resolution x-ray transmission images of the fast (ps scale) dynamics involved.

The main new science result of the paper is shown in Figures 2 and 3c. By combining time resolution with a variable photon probe energy, the authors are able to infer a temperature and ionization state vs space and time. There is not very much in the way of new physical insight and interpretation provided by the acquired data, though. Figure 3c (and Extended Data Figure 2) is hard to understand and learn something from given the lack of text describing what's happening. Is the variation from real time-dependent dynamics or variation in shots? The text implies 3b is related to 3c but it isn't clear based on the ~ 10 -20 values in the former and 2-4.5 values in the latter.

Figure 4 is used to justify the claim that the experiments have created "strongly coupled Fermi degenerate matter" in order to study it. Yet both the $\gamma=60$ and $\gamma=0$ points from the simulation (not data) are farther away from that regime than the starting point was. This comes off as a little strange and makes it seem that the Fermi degenerate claim was more of a buzzword use than quantitatively justified.

The point of Extended Data Figure 3d-3f is confusing and not clearly explained in terms of how it adds understanding to the experimental results. If there is something profoundly important discovered here, it hasn't been communicated to the reader very well.

The primary deficit of this manuscript as submitted is that it doesn't seem to meet the requirements for publishing in Nature. These types of lasers and targets have been actively studied for 20 years or more now. Now it has been done at SLAC with an XFEL diagnostic. This referee didn't find novel physical insight or new content in terms of how these measurements inform us about something we didn't know or expect, especially since the PIC code seems to have reproduced it (?), but rather this manuscript mostly just describes an experimental platform that will pave the way for more measurements in systems like it. As such, this referee cannot recommend publication in Nature or Nature Comm. but would instead suggest RSI or perhaps Phys Plasmas.

Reviewer #2 (Remarks to the Author):

The authors use an XFEL tuned near the Cu K-edge to visualise the propagation of the ionisation front in a 2- μm thick Cu target irradiated at $2 \times 10^{18} \text{ W/cm}^2$. They estimate the electron temperature from the relative change in the x-ray transmission. With the help of 2D PIC simulations, they attribute their experimental observations to electron impact ionisation by the energetic electrons propagating through the solid target without raising the electron temperature substantially. The research is topical, although there are several major issues with the manuscript.

For a 40-fs pump pulse, with a longitudinal dimension of $\sim 8.5 \mu\text{m}$ (considering an angle of incidence of 45 deg), irradiating a 2- μm foil, the rising edge of the main pulse should completely 'evaporate' the foil, and therefore at the peak intensity, the pulse should only 'see' an underdense plasma. Consequently, the initial conditions for the simulation setup (Extended Data Fig. 3a) are not particularly convincing. It is also not surprising that the authors did not observe various electron-heating mechanisms at play. However, given the poor signal-to-noise ratio of the experimental data

with 2- μm foils, would there be any detectable signal for thicker foils, where the hot electrons have a chance to propagate through an overdense plasma? Is it therefore justified for the authors to claim that this technique can disentangle the various competing heating and ionisation mechanisms?

More importantly, if the authors have created interaction conditions where the electrons propagate primarily through an underdense plasma, why do the authors need to resort to an expensive XFEL beamtime, since there are several well-established laser-plasma diagnostics to visualise ionisation fronts in underdense plasmas? I understand and appreciate the unique capabilities of XFELs, but they do not seem to be particularly relevant in this scenario.

In Fig. 3, the authors claim to present "electron temperature and mean charge state profiles averaged over the 2 μm thick solid Cu region". Firstly, as described above, the hot electrons see an underdense plasma, rather than a 'solid' Cu region. I hope that the authors have taken into consideration the different electron populations that are heated to different temperatures, and the cumulative effect of temperature and electron density, to compare with the change in the x-ray transmission. This should be clarified. Moreover, the authors firstly claim that the recirculating electrons increase the temperature of the peripheral region to 6-10 eV. This seems to be consistent with the temperature estimated from the experimentally measured change in the x-ray transmission in Fig. 2k. Yet, the discussion immediately shifts attention to "electron impact ionisation by the energetic electrons propagating in the solid target", which is presented as the sole mechanism responsible for the observed results. Why is the contribution of the recirculating electrons dismissed?

Fig. 2j is rather simplistic as the XANES spectrum is featureless. There exist detailed calculations in the literature - for instance, ref. 32 — that discuss the effect of non-equilibrated electron and ion temperatures on the XANES spectrum. I do appreciate that the approximate estimation of the upper and lower bounds on the electron temperature will perhaps not change appreciably, but this should be mentioned in the manuscript.

The authors mention, and rightly so, the ready accessibility of high-repetition-rate table-top petawatt lasers in contrast with single-shot lasers. Yet, it is not mentioned in the manuscript at what repetition rate the data was acquired. Did the authors utilise the full few-tens-of-Hz capability of SACLA? Was the target rastered? How was the probe-pump-probe sequence at each target position synchronised? Or was the experiment performed in a single-shot mode, in which case, the authors should refrain from advertising the capabilities of high-repetition-rate lasers for this particular experiment?

Although I appreciate the authors presenting near-raw experimental images, they should explain, in Fig. 1c and 1d, what the illuminated area beyond the right edge of the Cu foil ($x > 200 \mu\text{m}$) is. Is the XFEL spot size (not mentioned in the manuscript) larger than the target dimensions? In Fig. 1e, what causes the strong (presumably) noise along the left edge ($x < 200 \mu\text{m}$)? Is it due to pump scatter since the authors use a CMOS camera rather than an x-ray CCD camera? I would strongly recommend using a colour scheme that highlights the actual signal rather than the noise in all the transmission images throughout the manuscript.

In Fig. 3c, how are the different experimental data points at a given time-delay, but with different widths, retrieved? For example, at $\sim 0.3 \text{ ps}$, what do the six experimental data points at 9.05 keV with width varying from $\sim 50 \mu\text{m}$ to $\sim 120 \mu\text{m}$ correspond to? Are these shot-to-shot variations? I do agree with the authors that the experimental data points show qualitatively similar behaviour, compared to the simulations. However, given the wide range of experimental widths, it is only justified to put lower and upper bounds on $Z\text{-bar}$ as 2.0 and 4.5.

Extended data fig. 1f seems to have a very weak Fresnel diffraction pattern and it seems that the signal in Extended data fig. 1d-e might be buried in the very strong noise.

The manuscript should be proof-read. Do the authors mean a 'collimated' x-ray beam rather than

'parallel'. Repetitive phrases ("non-equilibrium strongly coupled Fermi degenerate matter") should be avoided in the abstract and "smeared edge profile" should be clarified as "smeared K-edge profile". There are a few other typographical errors in the manuscript.

On behalf of my co-authors, I would like to express our gratitude to both reviewers for their meticulous review and the insightful comments and suggestions provided for our manuscript submitted in 2022. In response to the feedback, we first published the technical details of our experimental technique in the Review of Scientific Instruments [Sawada, H. *et al.* “*Ultrafast time-resolved 2D imaging of laser-driven fast electron transport in solid density matter using an x-ray free electron laser*” Rev. Sci. Instrum. 94, 033511 (2023)]. We have since thoroughly revised our manuscript to address all the feedback received. We believe the revised version not only highlights our findings and significance of our work more clearly but also significantly enhances its readability for the laser plasma community and broader audiences.

We look forward to receiving your feedback on our revised manuscript.

Sincerely,
The Authors

To provide a detailed account of how we have addressed each point raised, our responses to the reviewers' comments are noted below in blue for clear differentiation.

Reviewer #1 (Remarks to the Author):

We appreciate the insightful comments and suggestions from Reviewer #1. We acknowledge that our previous manuscript did not effectively communicate our key findings. After detailing the technical aspects in RSI, we have substantially revised the introduction to clarify the current status of research and the existing challenges it faces. We now explicitly highlight our methods for spatiotemporal measurements using XFEL, detailing how these contribute to understanding solid target dynamics during high-intensity laser-plasma interactions.

Furthermore, we expanded the discussion on the implications of our work for fast ignition laser fusion. This method, utilizing high-intensity, short-pulse lasers to ignite a high-density fusion fuel, is a promising approach for advancing fusion energy production. It has garnered renewed attention following the achievement of scientific breakeven at the National Ignition Facility in December 2022. With our advanced spatiotemporal measurements, we have quantified a heating efficiency in a localized region of solid-density matter exceeding 100 eV within a picosecond. This capability of quantitative evaluation in heating efficiency paves the way for more efficient heating of high-density plasmas using state-of-the-art petawatt-class lasers. Such detailed measurements during the heating were not possible with traditional diagnostics such as picosecond X-ray streak cameras or 1D space-resolved X-ray spectrometers, which have limited resolution.

Additionally, our pump-probe measurements indicate an electron-ion energy relaxation time of tens of picoseconds, revealing the critical role of radiative cooling in the thermalization process. These measurements provide vital data on timescales that current numerical simulations cannot accurately predict.

Detailed responses for each comment are follows.

In this manuscript, the authors describe a series of experiments conducted with a short-pulse (fs scale) high-intensity ($2 \times 10^{18} \text{ W cm}^{-2}$) laser at SLAC. The experiments involved small copper targets that radiate characteristic emission near 9 keV where the K-edge is. The novelty in this work arises from the use of the XFEL beam available at SLAC as a diagnostic to provide high spatial and temporal resolution x-ray transmission images of the fast (ps scale) dynamics involved.

The main new science result of the paper is shown in Figures 2 and 3c. By combining time resolution with a variable photon probe energy, the authors are able to infer a temperature and ionization state vs space and time. There is not very much in the way of new physical insight and interpretation provided by the acquired data, though. Figure 3c (and Extended Data Figure 2) is hard to understand and learn something from given the lack of text describing what's happening. Is the variation from real time-dependent dynamics or variation in shots? The text implies 3b is related to 3c but it isn't clear based on the ~ 10 -20 values in the former and 2-4.5 values in the later.

As the reviewer pointed out, high-intensity laser-solid interactions have been studied extensively for over 20 years for fast ignition research. However, experimental measurements have traditionally been limited by diagnostics that integrate over space or time. For example, using time- and space-integrated X-ray spectroscopy, the electron temperatures of high-intensity laser-irradiated solid targets have been estimated to a range from a few eV and several keV. Although space-integrated, picosecond-time-resolved X-ray spectroscopy measurements suggest spatial gradients in such foils [e.g., Audebert, P. et al. Phys. Rev. Lett. 89, 265001 (2002)], spatiotemporal measurements with adequate resolutions have not been achieved, leaving uncertainties about which parts of the foil are heated and by which mechanisms.

In this work, we have overcome the challenge of spatiotemporal measurements by using femtosecond, collimated X-ray Free Electron Laser (XFEL) pulses at the SACLA XFEL facility in Japan. In cases of fast electron solid target heating, fast electrons transfer energy to the copper electrons through several heating and ionization mechanisms, illustrated in Figure 1a, while the target ions remain at the initial cold, solid density condition, known as isochoric heating. Thus, simultaneous measurements of the electron temperature and ionization state, rather than temperature and density, are critical to capture the rapidly evolving conditions in space and time. By using the tunability of the XFEL's photon energy, our measurements, shown in Figure 2, enabled us to determine that the electron temperature ranges between 7 and 18 eV. Moreover, Figures 3a and 3b, which illustrate simulations of the temporal evolution of the electron temperature and ionization state spatial profiles, are used to infer a range of ionization states. Our measurements in Figure 2 clearly show the transmission-affected area expands over time, agreeing more closely with the simulated ionization state than the electron temperature. Therefore, in Figure 3c, we compared the width of the expanding experimental feature with the simulated ionization state to estimate an ionization level ranging between 2 and 4.

We used the smearing of a K-edge profile to estimate the electron temperature of the solid foil. Although these represent only upper and lower limits, this quantitative estimate of temperature in fast electron-heated solid foils is demonstrated for the first time. Our spatiotemporal measurement

technique unambiguously identified a heated region, which cannot be measured with X-ray spectroscopy since the solid density matter at ~ 10 eV does not emit X-rays.

In Figure 3c, the relatively large spread of the data points at a given delay is due to shot-to-shot variations. We have revised the discussion of the causes of these variations in the last paragraph of the Results section.

Figure 4 is used to justify the claim that the experiments have created "strongly coupled Fermi degenerate matter" in order to study it. Yet both the $Y=60$ and $=0$ points from the simulation (not data) are farther away from that regime than the starting point was. This comes off as a little strange and makes it seem that the Fermi degenerate claim was more of a buzzword use than quantitatively justified.

We appreciate the reviewer's critical observation regarding our use of the term "strongly coupled Fermi degenerate matter." Upon reviewing this point, we recognize that the claim was not sufficiently supported by the data presented. Originally, we intended to indicate that the matter at $Y=60$ μm exhibits characteristics suggestive of Fermi degenerate matter, as evidenced by the smeared K-edge profile, placing it within the Fermi degenerate, warm dense matter. However, to ensure accuracy and clarity, we have removed the phrase from the current version of our manuscript and adjusted the description to more accurately reflect the characteristics observed.

The point of Extended Data Figure 3d-3f is confusing and not clearly explained in terms of how it adds understanding to the experimental results. If there is something profoundly important discovered here, it hasn't been communicated to the reader very well.

In the original manuscript, we presented 2D PIC simulations up to 1.5 ps, predicting a peak electron temperature of ~ 120 eV and an ionization state of ~ 20 , as shown in Figures 3a and 3b. These simulations suggested an electron-ion energy transfer rate of roughly 1 to 2 ps. However, this contrasts sharply with our experimental observation of diffraction fringes appearing at around 50 ps, indicating a notable discrepancy between the simulations and the experimental results.

We interpret this discrepancy as evidence of radiative cooling of the target, occurring between the end of the PIC simulation at 1.5 ps and the appearance of diffraction patterns at 50 ps. This observation indicates that dynamics on the tens of picosecond timescale, which are currently beyond the capability of PIC simulations, play a crucial role in the electron-ion thermalization process. We believe that radiative cooling significantly influences the energy transfer mechanisms that bridge our PIC simulation results with our experimental measurements.

To address this, Extended Data Figures 3d-3f in both the previous and revised manuscripts present calculations of the electron temperature and ionization state in solid copper as it undergoes radiative cooling. These calculations help estimate the target's temperature and ionization state throughout the period up to 50 ps, at which point a small localized region of the target is observed to ablate. This analysis provides critical insight into the dynamic cooling processes that reconcile the initially simulated conditions with the observed experimental results. Our revision aims to

clarify the connection between the simulated data and experimental findings, emphasizing the importance of radiative cooling.

The primary deficit of this manuscript as submitted is that it doesn't seem to meet the requirements for publishing in Nature. These types of lasers and targets have been actively studied for 20 years or more now. Now it has been done at SLAC with an XFEL diagnostic. This referee didn't find novel physical insight or new content in terms of how these measurements inform us about something we didn't know or expect, especially since the PIC code seems to have reproduced it (?), but rather this manuscript mostly just describes an experimental platform that will pave the way for more measurements in systems like it. As such, this referee cannot recommend publication in Nature or Nature Comm. but would instead suggest RSI or perhaps Phys Plasmas.

The spatiotemporally resolved measurements we conducted reveal two distinct plasma conditions in a solid-density target that is isochorically heated by laser-driven fast electrons: a highly charged hot solid-density plasma, and Fermi-degenerate warm dense matter, as summarized in Figure 4. While temperature-density contours describing warm dense matter and isochoric heating with short-pulse laser have been reported previously, such as “A Report on the SAUUL Workshop” by T. Ditmire and L. DiMauro (2002) and in Physics Reports by (10.1016/j.physrep.2018.04.001) by Dornheim et al. (2018), no experiments or simulations have shown the transition from solids to plasmas due to high-intensity laser-target interactions over the last two decades, as noted by the reviewer. Our results demonstrate this transition for the first time, making a significant advancement in illustrating solid-to-plasma transitions by fast electron isochoric heating.

Furthermore, while 2D PIC simulations have accurately captured our measurements up to ~ 1 ps, our measurements provide new evidence of a longer electron-ion energy transfer rate than predicted, addressing an open question in the field, and suggesting the critical role of radiative cooling. To the best of our knowledge, these experimentally based findings offer the first insights of this kind and are critical for understanding the dynamics of fast electron-heated solid target during both the heating and cooling phases.

We have revised the manuscript to first provide an overview of the current status of research and its challenges, followed by our approach to addressing these limitations. By publishing the details of the experimental technique used to capture the propagation of fast electrons in a separate RSI paper, this manuscript now focuses more on a detailed discussion of the physics involved and the implications of our findings. This shift highlights the novelty and importance of our work for fast ignition laser fusion, making it well suited for publication in in Nature Communications.

Reviewer #2 (Remarks to the Author):

We would like to express our appreciation to Reviewer #2 for their insightful comments and suggestions, particularly regarding the presentation of figures and the need for clarification on noise and intensity fluctuations in the data. We have addressed all the points raised in our revised manuscript. Our response to each comment is detailed below.

The authors use an XFEL tuned near the Cu K-edge to visualise the propagation of the ionisation front in a 2- μm thick Cu target irradiated at $2 \times 10^{18} \text{ W/cm}^2$. They estimate the electron temperature from the relative change in the x-ray transmission. With the help of 2D PIC simulations, they attribute their experimental observations to electron impact ionisation by the energetic electrons propagating through the solid target without raising the electron temperature substantially. The research is topical, although there are several major issues with the manuscript.

For a 40-fs pump pulse, with a longitudinal dimension of $\sim 8.5 \mu\text{m}$ (considering an angle of incidence of 45 deg), irradiating a 2- μm foil, the rising edge of the main pulse should completely ‘evaporate’ the foil, and therefore at the peak intensity, the pulse should only ‘see’ an underdense plasma. Consequently, the initial conditions for the simulation setup (Extended Data Fig. 3a) are not particularly convincing. It is also not surprising that the authors did not observe various electron-heating mechanisms at play. However, given the poor signal-to-noise ratio of the experimental data with 2- μm foils, would there be any detectable signal for thicker foils, where the hot electrons have a chance to propagate through an overdense plasma? Is it therefore justified for the authors to claim that this technique can disentangle the various competing heating and ionisation mechanisms?

As Reviewer #1 pointed out, the physics of high-intensity laser-solid interactions has been a key focus of study for over 20 years, largely driven by research on electron fast ignition. In the fast ignition concept for laser fusion, high-intensity, short-pulse lasers are used to ignite high-density fusion fuel. This rapid heating of a solid or high-density target by laser-driven fast electrons or protons is known as isochoric heating, where the density remains unchanged during the heating phase until hydrodynamic expansion occurs. Interest in fast ignition has recently been regained following the achievement of scientific breakeven at the National Ignition Facility in December 2022.

Isochoric heating of a solid density foil (thickness greater than $> 0.1 \mu\text{m}$) is a well-accepted concept in high-intensity, laser-solid interactions, even at an intensity of 10^{21} W/cm^2 . Instead of evaporating, the low-intensity pedestal or the rising edge of the peak laser pulse creates a thin layer of preplasma that shields the solid target from direct interaction with the high-intensity portion of the pulse. This preplasma facilitates the acceleration of free electrons to relativistic energies, primarily through the $\mathbf{J} \times \mathbf{B}$ force exerted by the intense laser field. The generation of fast electrons leads to the production of energetic protons through Target Normal Sheath Acceleration (TNSA). Experimentally, the persistence of solid targets after laser irradiation has been observed using monochromatic X-ray imaging, as reported in numerous literatures, such as Park, H. S. et al. (Phys. Plasmas 13, 056309, 2006) and our publication, Sawada, H. et al. (Phys. Rev. Lett. 122, 155002, 2019) or TNSA proton measurements by Patel, P. K. et al. (Phys. Rev. Lett. 91, 125004, 2003) and McKenna, P. et al. (Philos. Trans. R. Soc. A Math. Phys. Eng. Sci. 364, 711–723, 2006). Proton

acceleration does not occur if the target evaporates. Figure 1 of McKenna's paper illustrates the typical dynamics of laser-solid interactions.

Our experimental data also confirm the persistence of copper foils after laser irradiation. Figures 1c and 1d show raw X-ray images of the same copper foil, both with and without laser irradiation. The X-ray beam was collimated to a 1 mm diameter spot, which is larger than the 0.5 mm width of the copper strip. Both images clearly show that the 500 μm width of the copper foil remains intact 0.35 ps after the laser shot. If the foil had moved or evaporated, the X-ray transmission would have differed significantly between these pre- and post-laser shots.

Figure 1e shows the ratio of the images from Figures 1c and 1e. The spatial intensity of the X-ray pulses varies from shot to shot, a result of generating the X-ray pulses through self-amplified spontaneous emission. When calculating the division of images, the intensity variation between shots sometimes strongly persists, leading to significant gradients. To address the reviewer's concerns regarding the signal-to-noise ratio of the experimental data, we have revised the Method section to include an example analysis that estimates the threshold signal level, ensuring that our data used for the analysis is well above the noise level. In this experiment, we also used 10 μm and 25 μm thick copper foils. These thicker foils not only attenuate the X-ray intensities but also preserve the spatial structures of the foil surfaces in the transmitted images, as discussed in our RSI paper. These surface features make detecting the transmission changes we have described more challenging, which is why we presented only data for 2 μm thick copper foils in the manuscript.

More importantly, if the authors have created interaction conditions where the electrons propagate primarily through an underdense plasma, why do the authors need to resort to an expensive XFEL beamtime, since there are several well-established laser-plasma diagnostics to visualise ionisation fronts in underdense plasmas? I understand and appreciate the unique capabilities of XFELs, but they do not seem to be particularly relevant in this scenario.

We agree with the reviewer on the effectiveness of traditional optical imaging techniques for visualizing ionization fronts in underdense plasmas. However, the unique experimental conditions of our study necessitated the specific capabilities of the XFEL. Despite the high-intensity, short-pulse laser irradiation in our experiment, the target remains at its solid density due to isochoric heating, as detailed in our response to earlier comments.

In our experimental setup, at least three distinct plasma regions are produced within the intense-laser-irradiated foil: a high-temperature region roughly the size of the laser spot within the solid foil, surrounded by degenerate matter primarily driven by electron-impact ionization, and a low-density preplasma, as illustrated in Figure 1a. Hard X-ray pulses from the XFEL are critical because they can penetrate the preplasma and probe the underlying solid density regions, which are essential for our study. This capability is particularly valuable for investigating plasmas where temperatures are too low to emit X-rays.

We have revised our manuscript to further emphasize the importance of XFEL capabilities we used and the specific regions of solid-density copper foils to be probed in the third paragraph of the introduction.

In Fig. 3, the authors claim to present “electron temperature and mean charge state profiles averaged over the 2 μm thick solid Cu region”. Firstly, as described above, the hot electrons see an underdense plasma, rather than a ‘solid’ Cu region. I hope that the authors have taken into consideration the different electron populations that are heated to different temperatures, and the cumulative effect of temperature and electron density, to compare with the change in the x-ray transmission. This should be clarified. Moreover, the authors firstly claim that the recirculating electrons increase the temperature of the peripheral region to 6-10 eV. This seems to be consistent with the temperature estimated from the experimentally measured change in the x-ray transmission in Fig. 2k. Yet, the discussion immediately shifts attention to “electron impact ionisation by the energetic electrons propagating in the solid target”, which is presented as the sole mechanism responsible for the observed results. Why is the contribution of the recirculating electrons dismissed?

Regarding the role of recirculating electrons, they contribute to heating predominantly through Joule heating, while the impact of drag heating is negligible in solid density plasmas. Joule heating occurs when a beam of fast electrons propagates in a region where current neutrality has not been established, drawing a cold return current. Once the electrons begin recirculating around the target, both forward and backward currents are balanced by the fast electrons, making Joule heating ineffective. This phenomenon is supported by findings from literatures (e.g., Kemp, A. J., et al., Phys. Rev. Lett. 97, 235001 (2006)), our previous publication (Sawada H. et al, Phys. Rev. Lett. 122, 155002, 2019), and by the simulations presented in Figure 3a of the current manuscript.

We further elaborate on this point by detailing the physics of fast electron heating, including the negligible impact of drag heating under these conditions, in the fourth paragraph of the Results section of our manuscript.

Fig. 2j is rather simplistic as the XANES spectrum is featureless. There exist detailed calculations in the literature - for instance, ref. 32 — that discuss the effect of non-equilibrated electron and ion temperatures on the XANES spectrum. I do appreciate that the approximate estimation of the upper and lower bounds on the electron temperature will perhaps not change appreciably, but this should be mentioned in the manuscript.

Regarding the feature of the XANES spectrum, we consulted with Dr. S. X. Hu, an expert in theoretical and numerical research on atomic physics calculations, particularly first principles and Density Functional Theory modeling of high-density matter. His significant contributions are cited in our manuscript as Ref. 28 and 29. Based on his guidance, we revisited our analysis to calculate changes in the relative transmission ratios for each X-ray probe energy, and we have updated Figure 2e.

As anticipated, the revised temperature range inferred from this more detailed analysis is similar to our initial estimates. We have clearly indicated that these temperatures represent the upper and lower bounds of our measurements. Additionally, we have also added a few more references (Ref. 42 and 43) that elaborate on the use of smeared K-edge profiles as a model-free temperature diagnostic. These additions are detailed in the paragraph following Figure 2.

The authors mention, and rightly so, the ready accessibility of high-repetition-rate table-top petawatt lasers in contrast with single-shot lasers. Yet, it is not mentioned in the manuscript at what repetition rate the data was acquired. Did the authors utilise the full few-tens-of-Hz capability of SACLA? Was the target rastered? How was the probe-pump-probe sequence at each target position synchronised? Or was the experiment performed in a single-shot mode, in which case, the authors should refrain from advertising the capabilities of high-repetition-rate lasers for this particular experiment?

Our experiment was conducted on a single-shot basis. The synchronization of the XFEL and the optical laser beams, as well as their spatial overlap, was checked and adjusted each time the vacuum chamber was opened to exchange the set of target holders, which holds ~ 84 samples. The shot rate was approximately one shot per every three minutes. This rate was primarily limited by the need to move to a new target sample after each shot. Clearly, this does not constitute a high-repetition-rate experiment, as pointed out by the reviewer. Accordingly, we have removed the sentences on the capabilities of high-repetition-rate lasers for this particular experiment from the revised manuscript.

Although I appreciate the authors presenting near-raw experimental images, they should explain, in Fig. 1c and 1d, what the illuminated area beyond the right edge of the Cu foil ($x > 200 \mu\text{m}$) is. Is the XFEL spot size (not mentioned in the manuscript) larger than the target dimensions? In Fig. 1e, what causes the strong (presumably) noise along the leu edge ($x < 200 \mu\text{m}$)? Is it due to pump scatter since the authors use a CMOS camera rather than an x-ray CCD camera? I would strongly recommend using a colour scheme that highlights the actual signal rather than the noise in all the transmission images throughout the manuscript.

The XFEL spot size was $\sim 1 \text{ mm}$ in diameter, which is larger than the width of the copper strips. Although the beam was collimated, its spatial intensity varied from shot to shot due to the self-amplified spontaneous emission (SASE) mode. In response to the reviewer's question about the illuminated area beyond the right edge of the Cu foil ($x > 200 \mu\text{m}$) in Figures 1c and 1d where a magnification of 20 was used, it is important to note that the spatial intensity of an X-ray pulse extended beyond the width of the copper foil. This becomes more evident when X-ray images of a copper foil were recorded with a magnification of 10, which are presented in the RSI paper, as shown below. Each X-ray shot exhibits a slightly different intensity profile. Moreover, depending on the X-ray photon energy, the transmission through the $10 \mu\text{m}$ Cu foil varies significantly between 8.92 keV (below K-edge) and 9.05 keV (above K-edge). In X-ray only shots, we typically observe the spatially distribution of an X-ray pulse intensity attenuated by a solid copper foil and diffraction caused by the foil edges, as shown in the example images. The irradiation of the high-

intensity laser induces a change in transmission, which becomes pronounced after taking the ratio of two images between pre- and post-laser irradiation.

X-ray images of a 10 μm thick Cu foil recorded with a (a) 8.92 keV and (b) 9.05 keV probe.

We have added a description of the XFEL spot size and the source of intensity fluctuation in the first paragraph of the Results section. Additionally, we have changed the color scheme to better highlight the primary experimental features used in this analysis, as suggested by the reviewer, and added lineouts of the images. Their feedback has improved the presentations of our figures throughout the manuscript.

In Fig. 3c, how are the different experimental data points at a given time-delay, but with different widths, retrieved? For example, at ~0.3 ps, what do the six experimental data points at 9.05 keV with width varying from ~50 μm to ~120 μm correspond to? Are these shot-to-shot variations? I do agree with the authors that the experimental data points show qualitatively similar behaviour, compared to the simulations. However, given the wide range of experimental widths, it is only justified to put lower and upper bounds on $Z\text{-bar}$ as 2.0 and 4.5.

We appreciate the reviewer's comment regarding the reproducibility of the data. The scatter observed at a given time-delay is attributed to shot-to-shot variations. We collected data over several days and across multiple experimental runs, during which the vacuum chamber was opened to exchange target samples. These operations may cause slight variations in the relative timing between the XFEL beam and the optical laser. Furthermore, it is known that the relative timing tends to drift over times, as described in Ref. 38.

We have revisited our data analysis to determine if the statistics at each delay could be improved. Despite adding several data points, the relative spread of the data did not improve. Therefore, with the current dataset, we claim that our experimental data only set an upper and lower bound on the ionization state between $Z\text{-bar}$ values of 2.0 and 4.0.

We have added a description on the scatter of the experimental data in the section before the Discussion and in the Methods on page 13. Additionally, we have clarified the limits of the ionization state values.

Extended data fig. 1f seems to have a very weak Fresnel diffraction pattern and it seems that the signal in Extended data fig. 1d-e might be buried in the very strong noise.

We acknowledge the reviewer's observation regarding the weak diffraction pattern in Extended data Fig. 1f. Indeed, the signal in this particular example is extremely subtle. In addition to the weak signals, concentric patterns are affected by transmission changes, as shown in Figures 2c and 2f. Although only a limited number of X-ray images presenting concentric diffraction patterns well above noise levels were available, the qualitative analysis described in the Methods section enables us to identify an ablated area roughly the size of the laser spot ($\sim 26 \mu\text{m}$), rather than a 10 or 50 μm spot in diameter.

To measure the onset of the diffraction fringes, we used delay times of 20, 50, 100, 150 and 200 ps. The accuracy for determining the appearance of the diffraction pattern is only an upper bound of 50 ps. No diffraction patterns were observed at a 20 ps delay time.

The manuscript should be proof-read. Do the authors mean a 'collimated' x-ray beam rather than 'parallel'. Repetitive phrases ("non-equilibrium strongly coupled Fermi degenerate matter") should be avoided in the abstract and "smeared edge profile" should be clarified as "smeared K-edge profile". There are a few other typographical errors in the manuscript.

Thank you for your feedback. We have thoroughly proofread the revised manuscript to ensure clarity and accuracy. The term "parallel" has been corrected to "collimated" to more accurately describe the X-ray beam. We have also clarified "smeared edge profile" as "smeared K-edge profile" throughout the manuscript. Additionally, we have eliminated repetitive use of the phrase "non-equilibrium strongly coupled Fermi degenerate matter" in this version, and addressed other typographical errors identified in the manuscript.

REVIEWER COMMENTS

Reviewer #2 (Remarks to the Author):

The authors have satisfactorily answered all previous questions and the current manuscript, which seems to be largely rewritten, is a vast improvement on the previous version. The main thesis of the manuscript is the observation of a central core region (Region 1) in a 2- μm thick Cu foil illuminated by an intense laser at $2 \times 10^{18} \text{ W/cm}^2$. In this region, a temperature of 7-18 eV has been estimated from the variation in the transmission of an XFEL probe tuned to the Cu K-edge. The results are important and should be published, although they can still be better communicated. For instance, in the introduction, the authors write in detail about various heating mechanisms and how the novel use of the XFEL has been exploited in their experiment to spatio-temporally resolve the electron temperature and the mean ionization state, and how this technique is a significant improvement on conventional spatio-temporally integrated x-ray emission data. Yet, in the very next paragraph ("Here, we present..."), the authors infer a blanket temperature of 7-18 eV, and a blanket ionization state of 2-4, which is neither spatially nor temporally resolved information. One therefore wonders whether this information could have been easily inferred using other diagnostics instead of an expensive XFEL beamtime, as the other referee also pointed out. For example, can the authors comment on the temperature of the surrounding Region 2, or is it below the threshold of detection? Similarly, can the authors estimate the temperature at 0.4 ps and 1.0 ps (if not 150 ps due to the diffraction rings) from Fig. 2a-i? The only space-time resolved data is Fig. 3c, despite the large experimental error bars. Can the authors make an equivalent plot for electron temperature, since that is the very claim at the introduction of the manuscript? Overall, the experimental data is of interest and has potential, and the current version of the manuscript is significantly more cohesive compared to the previous version. However, further work needs to be done to effectively bring out the spatio-temporally resolved nature of the experimental data to support the claims of novelty at the introduction.

Minor points:

The description of figure 2 in the text (Page 4) mentions a time-delay of 0.3 ps, whereas the figure shows data at 0.4 ps.

It would have been helpful to specify which Methods section to refer to in the main text, so that relevant sections could be read without having to go through the entire Methods description.

Reviewer #3 (Remarks to the Author):

The manuscript by Sawada and co-workers presents the use of an XFEL (specifically the SACLA facility) to study fast electron transport driven by an ultra-intense short-pulse laser.

The main value in this manuscript is presenting the first such study. In this respect there is a high degree of novelty, and this adds to current knowledge. It also has some value in demonstrating the continuing validation of fast electron transport codes. That said, looking at fig. 3, one should probably stop short of saying that the agreement is 'very good', or rather that the experimental measurements are really sufficient to provide such a test.

In terms of the insights into the physics of fast electron transport however, this manuscript does not add too much. Much of the physical phenomena that are described in this paper have already been acknowledged in the current literature. This paper does not, in the view of this referee, either challenge or add much to the core physical understanding of fast electron transport.

Considering the merits and downsides of this paper, I believe that it should be accepted into Nature Communications. The primary benefit of this paper is to demonstrate experimental progress in probing

fast electron transport and should provide a 'confidence shift' in the current abilities to model and understand fast electron transport.

Point-by-point response to the Reviewers' comments (by H. Sawada et al., NCOMMS-22-09423A-Z) - 6/29/2024

On behalf of the co-authors, I would like to express our gratitude to both reviewers for recognizing the importance of our work and providing insightful comments. In response to their feedback, we have thoroughly revised our manuscript to address all the feedback received.

We believe the revised version not only clarifies our findings and emphasize the significance of our work but also significantly enhances its readability for experts in laser plasma community and broader audiences.

We look forward to receiving your feedback on our revised manuscript.

Sincerely,
The Authors

To provide a detailed account of how we have addressed each point raised, our responses to the reviewers' comments are noted below in blue for clear differentiation.

REVIEWER COMMENTS

Reviewer #2 (Remarks to the Author):

We would like to express our appreciation to Reviewer #2 for their constructive comments and for recognizing the improvements in our manuscript. We have carefully addressed all the points raised and made the necessary changes, which are highlighted in the revised manuscript. Our detailed responses to each comment are provided below.

The authors have satisfactorily answered all previous questions and the current manuscript, which seems to be largely rewritten, is a vast improvement on the previous version. The main thesis of the manuscript is the observation of a central core region (Region 1) in a 2-um thick Cu foil illuminated by an intense laser at 2×10^{18} W/cm². In this region, a temperature of 7-18 eV has been estimated from the variation in the transmission of an XFEL probe tuned to the Cu K-edge. The results are important and should be published, although they can still be better communicated. For instance, in the introduction, the authors write in detail about various heating mechanisms and how the novel use of the XFEL has been exploited in their experiment to spatio-temporally resolve the electron temperature and the mean ionization state, and how this technique is a significant improvement on conventional spatio-temporally integrated x-ray emission data. Yet, in the very next paragraph (“Here, we present...”), the authors infer a blanket temperature of 7-18 eV, and a blanket ionization state of 2-4, which is neither spatially nor temporally resolved information.

A thin copper foil irradiated by a high-intensity, short-pulse laser exhibits at least three different plasma regimes: a hot, solid-density region near the laser focal spot (Region 1), a peripheral colder, solid-density region (Region2), and a low-density, hot preplasma, as illustrated in the introduction

and in Figure 1a. The inferred range of electron temperatures (7~18 eV) and ionization states (2~4) pertains specifically to Region 2, which is spatially distinct from those in Region 1 and preplasma.

Regarding the temporal evolution, the electron temperature in Region 2 is quickly heated to several eV in ~0.1 ps and remains constant afterward, as shown by the simulations in Figures 3a and 4. This explains why the electron temperature does not increase over time after ~0.1 ps. Meanwhile, the inferred ionization state appears constant because it corresponds to conditions behind the electron heat front, which expands over time. Our measurements demonstrate the propagation of the heat front, as shown in Figures 2a, 2b, 2d and 2e. Thus, the experimentally inferred ionization state is not fixed at a specific spatial position.

Our experimental data are indeed spatiotemporally resolved. However, as the reviewer pointed out, the inferred electron temperature and ionization states do not exhibit both temporal and spatial dependence. To clarify this, we have revised the third paragraph of the introduction as follows:

“Here, we present the investigation of the spatiotemporal dynamics of laser-driven fast electron heating in a solid-density copper foil, achieving sub-micron and femtosecond resolutions using an X-ray Free Electron Laser (XFEL).”

This revision clarifies that our work reveals the spatiotemporal dynamics of fast electron heating, rather than simply claiming spatially and temporally resolved measurements of T_e and \bar{Z} within the solid-density, heated target, which might have been misleading.

One therefore wonders whether this information could have been easily inferred using other diagnostics instead of an expensive XFEL beamtime, as the other referee also pointed out. For example, can the authors comment on the temperature of the surrounding Region 2, or is it below the threshold of detection? Similarly, can the authors estimate the temperature at 0.4 ps and 1.0 ps (if not 150 ps due to the diffraction rings) from Fig. 2a-i? The only space-time resolved data is Fig. 3c, despite the large experimental error bars. Can the authors make an equivalent plot for electron temperature, since that is the very claim at the introduction of the manuscript?

Diagnostics to infer plasma conditions of 7-18 eV and above solid density, which fall within the warm dense matter regime, are severely limited. The technique we developed is novel and has not been recognized in the literature yet (e.g., K. Falk, “Experimental methods for warm dense matter research”. *High Power Laser Science and Engineering*. 2018;6:e59.). The plasma temperature of warm dense matter is too low to emit X-rays for X-ray emission spectroscopy, and the high-density plasma self-absorbs extreme ultraviolet radiation. Thus, probing such matter requires an external bright X-ray source, such as those from laser-produced plasmas or XFELs. Our manuscript presents the first application of XFEL-based transmission imaging to diagnose fast electron-heated warm dense copper. In particular, the spatially resolved information of heating in Cu (Figures 2a-2i) is impactful because no previous measurements have captured the expansion of the heated region with sub-picosecond and a micron-scale resolution.

The temperature in Region 2 is inferred to be 7-18 eV. In contrast, the temperature and ionization state in Region 1 are around 120 eV and 20, respectively, at ~1.5 ps, according to PIC simulations.

For this region, the smearing of the X-ray absorption spectrum near the K-edge was not applicable because T_e is greater than T_F for Region 1.

At 0.4 ps and 1.0 ps, the temperature in Region 2 is around several eV, based on simulations. As described above, the electron temperature in Region 2 is heated to this level in ~ 0.1 ps and only gradually increases over time afterward. Our inferred range of temperature (7~18 eV) is larger than its variation.

Figure 4 summarizes the temporal and spatial evolution of plasma conditions for Regions 1 and 2. The solid lines representing the simulation results show the transition from cold solid copper to heated copper from 0 ps to ~ 1.5 ps, while the white box indicates the experimentally inferred condition. We have added the label of Region 1 and Region 2 to the figure to clarify that the experimental condition is close to those in Region 2. As indicated by a dot on the solid red line at 0.1 ps in the graph, the electron temperature in Region 2 increases so quickly that the current temporal resolution of 0.1 ps is insufficient to resolve the initial temperature rise.

To ensure consistency between our claim in the introduction and our data, we have revised the introduction as mentioned before. Additionally, we have added discussions on the inapplicability of the technique for Region 1 and the requirement of a faster temporal resolution to capture the initial rise in the electron temperature to the first paragraph of Discussion.

Overall, the experimental data is of interest and has potential, and the current version of the manuscript is significantly more cohesive compared to the previous version. However, further work needs to be done to effectively bring out the spatio-temporally resolved nature of the experimental data to support the claims of novelty at the introduction.

We have clarified the reviewers' concerns and addressed all the comments raised in this response and revised manuscript. In the introduction, we have made necessary revisions to clarify that our measurements probed only Region 2, where solid copper is heated by an expanding fast electron heat front, and that we did not show a temporal evolution. Additionally, we have clarified that the spatiotemporal information is derived from a combination of our spatiotemporal measurements and simulations, as highlighted in the introduction. Specifically, these measurements include those with the diffraction fringes that we used to infer the electron-ion energy relaxation time and the ablated region.

Minor points:

The description of figure 2 in the text (Page 4) mentions a time-delay of 0.3 ps, whereas the figure shows data at 0.4 ps.

The time delay mentioned in the text has been corrected from ~ 0.3 ps to ~ 0.4 ps to match the data shown in Figure 2.

It would have been helpful to specify which Methods section to refer to in the main text, so that relevant sections could be read without having to go through the entire Methods description.

We thank the reviewer for their suggestion. In response, we have revised the section names in the Methods and added these specific names in the main text. This allows readers to easily find the relevant Methods section as they read through the main text.

Reviewer #3 (Remarks to the Author):

We would like to express our appreciation to Reviewer #3 for recognizing the importance of our research work and providing insightful comments. Although these are mainly comments rather than requests for revisions, we have responded to each comment in detail below and have made necessary changes to improve readability.

The manuscript by Sawada and co-workers presents the use of an XFEL (specifically the SACLA facility) to study fast electron transport driven by an ultra-intense short-pulse laser.

The main value in this manuscript is presenting the first such study. In this respect there is a high degree of novelty, and this adds to current knowledge. It also has some value in demonstrating the continuing validation of fast electron transport codes. That said, looking at fig. 3, one should probably stop short of saying that the agreement is 'very good', or rather that the experimental measurements are really sufficient to provide such a test.

We appreciate your recognition of the novelty and value of our study. The inferred plasma conditions of the electron-heated copper foil, specifically in Region 2, fall within the warm dense matter regime. Our measurement technique is a novel diagnostic for such matter, and we will elaborate on this in response to the comments below.

As the reviewer pointed out, the experimental measurements shown in Figure 3c can be explained by simulations with Z of 2~4, which is insufficient to provide a benchmark test. Additionally, the experimental measurements are only for Region 2. Plasma conditions of the hot, solid-density plasmas in Region 1 need to be measured separately for complete benchmarking. This point is emphasized in the first paragraph of the discussion section.

In terms of the insights into the physics of fast electron transport however, this manuscript does not add too much. Much of the physical phenomena that are described in this paper have already been acknowledged in the current literature. This paper does not, in the view of this referee, either challenge or add much to the core physical understanding of fast electron transport.

The physical heating mechanisms, such as Joule heating, drag heating, and diffusive heating, have been acknowledged in the literature, as the reviewer pointed out. However, we would like to emphasize that successfully inferring the plasma condition in Region 2 is significant. As described in the introduction, the three heating mechanisms are difficult to disentangle. Our spatially and temporally resolved measurements enabled us to distinguish the contribution of those mechanisms. In particular, the increase in electron temperature and ionization state in Region 2 is attributed to Joule heating and electron impact ionization, respectively, as illustrated in Figure 1a. The inferred electron temperature (7~18 eV) and ionization state (2~4) clearly benchmark the heating and ionization physics incorporated in the simulation code. Furthermore, determining the ionization state for warm dense matter of mid- Z materials is challenging due to the effects of continuum lowering (ionization potential depression). Combining our XFEL measurements and PIC simulations could be a new means to investigate such physics.

To clarify what physical understanding of fast electron transport has been revealed, we have added the above discussion to the first paragraph of Discussion. We have also commented on the potential

application of our measurement technique to investigate dense plasma effects in the second paragraph of the Discussion section.

Considering the merits and downsides of this paper, I believe that it should be accepted into Nature Communications. The primary benefit of this paper is to demonstrate experimental progress in probing fast electron transport and should provide a 'confidence shift' in the current abilities to model and understand fast electron transport.

Again, we appreciate the reviewer for recognizing the importance of work. We believe that the changes we have made in response to the reviewer's comments have improved readability and clarifies our findings better than in the previous manuscript.